# Identifying Missing Sources and Reducing NO$_x$ Emissions Uncertainty over China using Daily Satellite Data and a Mass-Conserving Method

Lingxiao Lu[1], Jason Blake Cohen[1*], Kai Qin[1*], Xiaolu Li[1], Qin He[1]

[1]Shanxi Key Laboratory of Environmental Remote Sensing Applications, China University of Mining and Technology, Xuzhou, 221116, China

*Correspondence to*: Jason B.Cohen ([jasonbc@alum.mit.edu](mailto:jasonbc@alum.mit.edu) ; [jasonbc@cumt.edu.cn](mailto:jasonbc@cumt.edu.cn) ; )

**Abstract.** This study applies a mass-conserving model-free analytical approach to daily observations on a grid-by-grid basis of NO$_2$ from TROPOMI, to rapidly and flexibly quantify changing and emerging sources of NO$_x$ emissions at high spatial and daily temporal resolution. The inverted NO$_x$ emissions and optimized underlying ranges include quantification of the underlying atmospheric in-situ processing, transport and physics. The results are presented over three changing regions in China, including Shandong and Hubei which are rapidly urbanizing and not frequently addressed in the global literature. The day-to-day and grid-by-grid emissions are found to be 1.96±0.27 µg/m$^2$/s on pixels with available priori values (1.94 µg/m$^2$/s), while 1.22±0.63 µg/m$^2$/s extra emissions are found on pixels in which the a priori inventory is lower than 0.3 µg/m$^2$/s. Source attribution based on thermodynamics of combustion temperature, atmospheric transport, and in-situ atmospheric processing successfully identify 5 different industrial source types. Emissions from these industrial sites adjacent to the Yangtze River are found to be 161.±68.9 Kt/yr (163% higher than the a priori) consistent with missing light and medium industry located along the river, contradicting previous studies attributing the water as the source of NO$_x$ emissions. Finally, the results demonstrate those pixels with an uncertainty larger than day-to-day variability, providing quantitative information for placement of future monitoring stations. It is hoped that these findings will drive a new approach to top-down emissions estimates, in which emissions are quantified and updated continuously based consistently on remotely sensed measurements

and associated uncertainties that actively reflect land-use changes and quantify misidentified emissions, while quantifying new datasets to inform the bottom-up emissions community.

**1 Introduction**

The sum of Nitrogen Oxide (NO) and Nitrogen Dioxide ($NO_2$) hereafter called $NO_x$, is produced during fossil fuel, biomass, and other combustion process or heat sources due to the re-combination of atmospheric $N_2$ and $O_2$ (Brewer et al., 1973; Logan, 1983). $NO_x$ is a short-lived trace gas that directly impacts health, nitrate aerosol, tropospheric ozone (both an air pollutant and greenhouse gas), and the OH radical, which indirectly impacts both CO and $CH_4$ (Alcamo et al., 1995; Chen et al., 2007; Collins et al., 2013; Crutzen, 1970; Jacob et al., 1996; Li et al., 2018; Monks et al., 2015; Prather, 1996; Rigby et al., 2017; Rollins et al., 2012; Sand et al., 2016; Seinfeld, 1989; Shindell et al., 2012; Tan et al., 2018). Although there are techniques to observe in-situ surface $NO_x$ and the atmospheric column of $NO_2$ during the daytime via remote sensing in the UV and visible portions of the spectrum, there is no way to observe the atmospheric burden of column $NO_x$, while even observations of $NO_2$ during the night time are not reliable (Bauwens et al., 2020; Bechle et al., 2013; Boersma et al., 2009; Lamsal et al., 2014; Lee et al., 2014; Russell et al., 2011; Van Geffen et al., 2020). Furthermore, due to rapid atmospheric chemistry, interactions with UV radiation, sensitivity to temperature and vertical structure, and pseud-steady state balance between $NO_2$ and NO, there is no simple way to quantify rapidly changing or emerging sources of $NO_x$ emissions at high spatial and daily temporal resolution (Alvarado et al., 2010; Leue et al., 2001; Martin et al., 2003, 2006; Mijling et al., 2013).

Present approximations of $NO_x$ emissions tend to miss emerging sources and underestimate sources undergoing rapid change, while also overestimating highly regulated sources, leading to a combination of biases, as well as being rapidly outdated compared to rapid changes in the emissions and economic structure, particularly so in the Global South (Cohen and Wang, 2014; Dados and Connell, 2012; Lin et al., 2020; Qin et al., 2023; Wang et al., 2021; Zhu et al., 2014). Bottom-up

aggregation uses a small subset of spatial and temporal measurements in the field or sometimes in the laboratory, and combines
these with economic, technological, and other data to scale up emissions (Amstel et al., 1999; Bond et al., 2004; European
Commission. Joint Research Centre., 2021; Li et al., 2017; Olivier et al., 1994; Oreggioni et al., 2021). This approach can also
be applied to biomass burning emissions by including fire radiative power and other indirect remotely sensed measurements
of land-use change, which are then scaled based on small spatial and temporal measurements of emissions factors, biomass,
and other available data (Cohen et al., 2017; Giglio et al., 2013; Van Der Werf et al., 2017; Wang et al., 2020). Direct flux
measurements can be made via a sparse network of local flux towers, each with a limited spatial range and operating under
standard meteorological conditions (Geddes and Murphy, 2014; Haszpra et al., 2018; Karl et al., 2017; Lee et al., 2015).
Chemical transport models can be merged with Bayesian, Data Assimilation, or Kalman Filter approaches to invert emissions
and produce error estimates, which in turn consume a huge amount of computational time and requiring explicit knowledge of
the errors of every input variable, including those in the modeling system itself (Cohen and Wang, 2014; Henderson et al.,
2012; Napelenok et al., 2008). There have even been some direct inversions of results from isolated and very strong, non-time
varying sources, requiring that these sources be surrounded by clean background conditions, and applying the very strict
assumptions of Gaussian plume modeling (Beirle et al., 2011, 2019; Cohen and Prinn, 2011; De Foy et al., 2014; Jin et al.,
2021; Laughner and Cohen, 2019), or by integrating data over a long and continuous period of time, over a specific season or
other set of conditions which is generally not changing, and then assuming fitting the average spatial and temporal emissions
(Kong et al., 2022).

60        Although the above methods have their own advantages, there are still significant problems including: missing of sources,

underapproximation of small and moderate sources (Beirle et al., 2021; Drysdale et al., 2022; Qin et al., 2023), underestimation
of the spatial and temporal variability of sources with large variability (Stavrakou et al., 2016; Vaughan et al., 2016; Wang et
al., 2010; Zyrichidou et al., 2015), and the inability to scale a priori regions with zero emissions (Cohen, 2014; Zhao and Wang,
2009). In general, these methods do not provide an uncertainly analysis, or require model and measurement uncertainty to be
highly parametrized (Bond et al., 2013; Cohen and Wang, 2014). There is no reason why $NO_x$ emissions should be static in
time, or should have a constant ratio of NO to $NO_2$, even though these are current assumptions which are built into most models
used by the community (Li et al., 2023b). This combination of weaknesses has limited most emissions studies to scaling-based
perturbations of $NO_x$ emissions, without considering the spatial and temporal variation in the distribution, therefore requiring
the implicit adaptation of large spatial and temporal averages (Evangeliou et al., 2018; Lund et al., 2020; Wang et al., 2021).
This in turn tends to miss significant emissions sources from rapidly changing sources such as wildfires, missed sources
including new urbanization, and sources which are changing due to changes in the climate system itself (Deng et al., 2021).
This work applies the recently introduced mass conserving model free approximation of $NO_x$ emissions [MCMFE-$NO_x$]
approach (Li et al., 2023a; Qin et al., 2023), using daily-scale remotely sensed tropospheric columns of $NO_2$ from TROPOMI
at 3.5km×7/5.5km spatial resolution in combination with reanalysis wind fields to approximate the daily $NO_x$ emissions and
uncertainty ranges over major population and economic regions of Greater China. The specific results herein are applied to
robustly account for the uncertainties in the remotely sensed column observations of $NO_2$, actively provide a quantification of
the range of thermodynamics driving the ratio of the NO to $NO_2$, dynamical transport, and a first order in-situ chemical loss,
all within the context of the tropospheric column measurement and a priori emissions uncertainty ranges. This approach allows
for non-linear feedbacks to be accounted for, including those from climate-induced changes to policy induced changes, some
of which are analyzed in the context of the results provided. The modeling was done on a PC and is model-independent,
allowing the results to be rapidly reproduced, or improved upon with updated measurements, physical, chemical, and other
routines, and to be integrated rapidly into all existing modeling and policy frameworks, with little to no additional
computational cost (Cohen et al., 2011; Cohen and Prinn, 2011; Holmes et al., 2013; Prinn, 2013).
In this work, MCMFE-NO$_x$ is applied over three rapidly changing regions (Figure 1) in China with densely urbanized
sub-regions, and surrounding rural, rapidly developing suburban and urbanizing sub-regions, and new development aiming to
upgrade the urban areas and energy intensive industries in these areas to meet the large-scale developmental and climate goals
set by the Chinese National government (Bao, 2018). The detailed emissions estimates are made using one year of daily
TROPOMI NO$_2$ data. Unlike the vast majority of air pollution emissions studies which focus on the three large and well
characterized locations of the Beijing-Tianjin-Hebei, Yangtze River Delta, and the Pearl River Delta (Haas and Ban, 2014;
Wang et al., 2022; Yang et al., 2021), the estimates specifically include adjacent areas which include large cities with overall
populations similar to or larger than the previously studies areas, specifically including: Wuhan along the middle Yangtze
River, Qingdao, Jinan, and others in Shandong Province, and Shantou and Xiamen along the South China Sea. In addition,
rapidly industrializing locations such as Zibo, Ma'anshan, and Beihai are included, which were previously not included. The
estimates also include highly developed cities such as Beijing, Shanghai, and Hong Kong (which has never had a gridded a
priori emissions developed in the past by either MEIC or EDGAR), cover cities which have recently reached highly developed
status but have undergone a large amount of recent change including Nanjing, Suzhou, Dongguan, and Foshan, heavily coal-
based and oil-based resource regions such as Tianjin, and Tangshan, industrial cities including Xuzhou, and agricultural areas
such as Jining, Heze, Meizhou, and Xinyang (Cai et al., 2019; Chang and Kim, 1994; Dhakal, 2009; Liu et al., 2021; Wu,
2016; Zhang et al., 2008; Zhuang et al., 2022). The large amount of variability of sources, rapid economic development, and
strong changes in environmental emissions policy and regulation, have led to significant changes in terms of emissions
magnitude, in both space and time over this region (Carson et al., 1997; Charfeddine and Kahia, 2019). Traditionally, NO$_x$
emissions from water bodies have been regarded as negligible. Some findings have reported that NO$_x$ emission from lakes is
due to several biological and microbial processes (Kong et al., 2023). Other findings have only considered that the contribution
of NO$_x$ emissions over water must be attributed to shipping activities (Zhang et al., 2023). However, in this work, NO$_x$

emissions and underlying forcing properties inverted day-by-day at a 0.05° x 0.05° grid resolution clearly point to the fact that there are missing small and medium-sized power plants and industrial facilities which play an essential role and can also produce significant emissions, which these other studies have presently overlooked. The co-location along the edges of water bodies is in part due to the fact that these sites can both use the water cooling purposes as well as to possibly transport incoming and/or outgoing raw material and products.

## 2 Data and methods

### 2.1 Geographic Boundaries of Study Region

In the realm of published air pollution research in China, most scholarly work has concentrated on three different regions: Beijing and surrounding area, Shanghai and surrounding area, and Guangzhou and surrounding area. The first of these regions is usually defined as encompassing Beijing, Tianjin, and Hebei. In this work, we have instead opted to take a view based on the column loading climatology of $NO_2$ as well as industrial and population density, as displayed in Figure 1. First, since it is observed that $NO_2$ loadings in Hebei and near the great wall in northern Beijing are relatively low north of 40.5°, this work places a boundary here. Other regions are identified in which the column $NO_2$ has a climatology smaller than $1.43 \times 10^{15}$ molec/cm$^2$ are also excluded. The goal is to delineate a boundary along a contiguous contour of high $NO_2$ climatological loading, implying that the data needed to compute the emissions will be more clear and less influenced by observational noise. Our proposed continuous region 1 encompasses a substantial portion of adjacent Shandong province to the south and east, which is known to have both a high population density and extensive mineral, oil, and heavy manufacturing enterprises. The eastward extent ends in Qingdao (with a population of 9.5 million people and considerable manufacturing and ports). To the south, the region extends into far northern Jiangsu province, encompassing the cities of Xuzhou (with a population of 8.8 million people and considerable moderately intensive industry) and Lianyungang (one of the largest ports in China). We have

used a similar approach to extend the region commonly used around Shanghai to match with the observed climatological
loadings of $NO_2$. The new area extends up the Yangtze river far to the west, and now includes the city of Wuhan (a population
of more than 9.1 million and growing, and considerable industry). Additionally, there are new locations in between which are
identified and included which are characterized by burgeoning coal utilization or energy infrastructure, as well as rapid
population and industrial development. The region has nearly doubled/tripled in size by including the continuous region west
from Nanjing and Hangzhou all the way through Wuhan, as displayed in Figure 1. Similarly the typical regions in the south
have been extended beyond the Guangzhou to Shenzhen and other adjacent cities in the pearl river delta. The new region
includes substantial urban, financial, and commercial centers such as Hong Kong and Xiamen, which are previously excluded.
Similarly, other industrial cities and port cities such as Beihai and Shantou are also included. These cities, now stretching along
the South China Sea continuously from the Vietnamese border to the East China Sea, provide a broader perspective on the
geographical scope of our research, and account for the unique characterisitcs of the Asian Monsoon in a consistent manner
(Cohen, 2014; Ding et al., 2021; Wang et al., 2021).

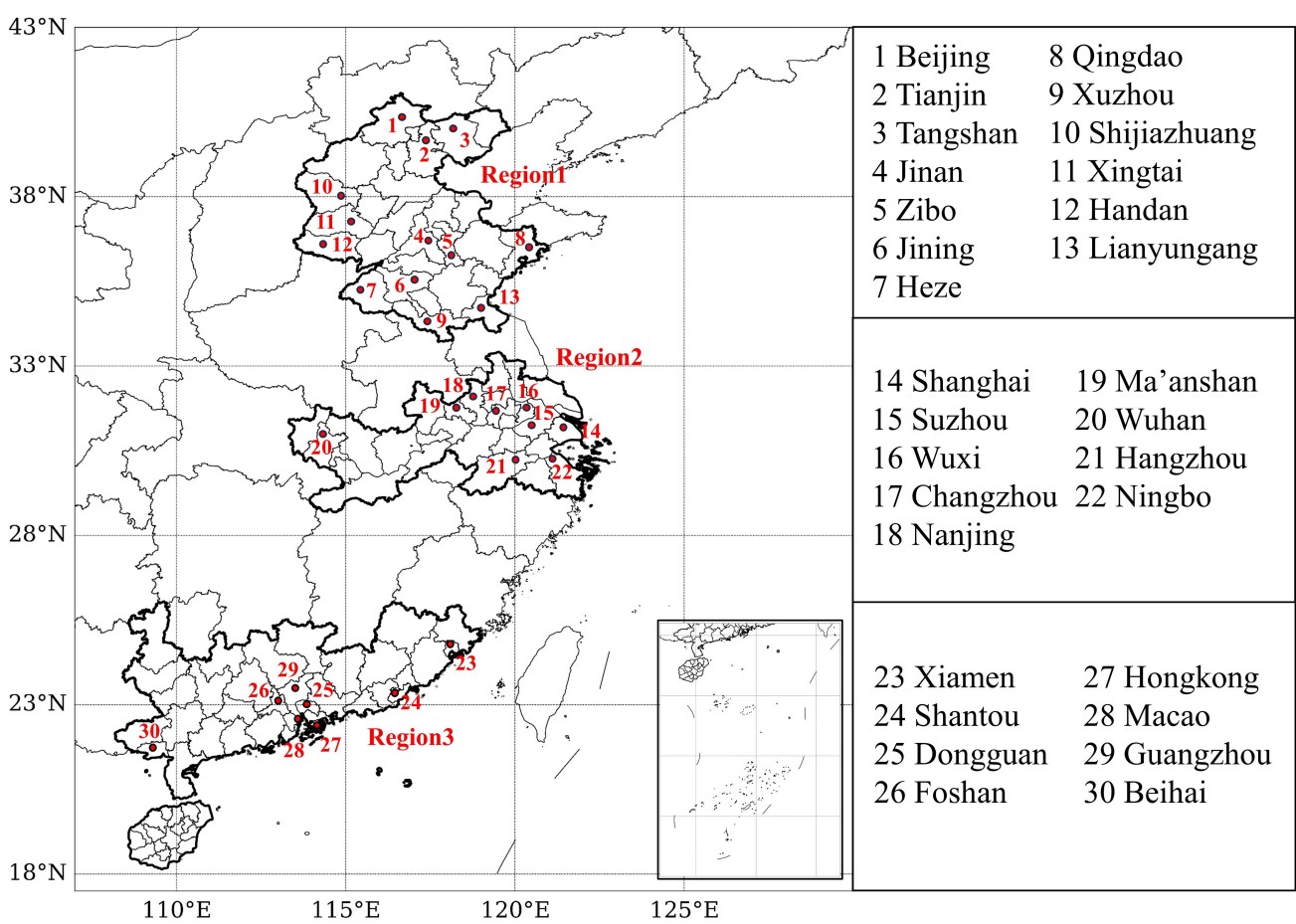

| 1 Beijing | 8 Qingdao |
| 2 Tianjin | 9 Xuzhou |
| 3 Tangshan | 10 Shijiazhuang |
| 4 Jinan | 11 Xingtai |
| 5 Zibo | 12 Handan |
| 6 Jining | 13 Lianyungang |
| 7 Heze | |

| 14 Shanghai | 19 Ma'anshan |
| 15 Suzhou | 20 Wuhan |
| 16 Wuxi | 21 Hangzhou |
| 17 Changzhou | 22 Ningbo |
| 18 Nanjing | |

| 23 Xiamen | 27 Hongkong |
| 24 Shantou | 28 Macao |
| 25 Dongguan | 29 Guangzhou |
| 26 Foshan | 30 Beihai |

Figure 1: A map of the three study regions, including names and locations of 30 important cities mentioned in this work.

## 2.2 TROPOMI Tropospheric NO$_2$ Column Retrievals

The Sentinel-5 Precursor satellite from the European Space Agency, is equipped with an advanced instrument known as the Tropospheric Monitoring Instrument (TROPOMI) (Van Geffen et al., 2020; Veefkind et al., 2012), which is a nadir-viewing spectrometer with an overpass time of approximately 13:30 local solar time. The TROPOMI spectrometer measures ultraviolet (UV), visible and near-infrared spectral bands, allows observation of NO$_2$ as well as other air pollutants, aerosols and clouds. TROPOMI measures NO$_2$ vertical columns with a spatial resolution of 3.5 x 7 km$^2$ (reduced to 3.5x5.5 km$^2$ since August 2019) and with a swath width of ~2600 km.

The research herein uses the reprocessed dataset S5P-PAL, Version 2.3.1, and includes all days with data from 1st January
2019 to 31st December 2019. The selection of the year 2019 is based on its status as the first complete year of NO$_2$ retrievals
by Sentinel-5P. To ensure data quality, only pixels with a "qa_value" of 0.75 or higher are utilized. This pixel filter, which is
recommended for most users, excludes cloud-covered scenes (cloud radiance fraction > 0.5), portions of scenes covered by
snow or ice, errors, and problematic retrievals (https://data-portal.s5p-pal.com) (Van Geffen et al., 2022). As shown in Figures
2a and 2b, the pixels of NO$_2$ column observations within each swath are amalgamated into unified latitude-longitude grids
measuring 0.05°x0.05° in size, using the weighted polygon shaped remotely sensed measurement toolkit HARP
(http://stcorp.github.io/harp/doc/html/index.html). An area weighted average is performed, ensuring that the re-gridded values
accurately        represent        the        spatial        distribution        of        the        original        data
(http://stcorp.github.io/harp/doc/html/algorithms/regridding.html).
**2.3 Prior Emissions Inventory**
The assumptions regarding NO$_x$ emission datasets in the initial step applied are harmonized using multi-source
heterogenous data, developed by the MEIC (Multi-resolution Emission Inventory for China) team (Huang et al., 2012, 2021;
Kang et al., 2016; Liu et al., 2016; Zheng et al., 2021; Zhou et al., 2017, 2021), in collaboration with various scientific research
institutions. This dataset is referred to as the high-resolution INTegrated emission inventory of Air pollutants for China
(INTAC), which is highlighted in purple in Figure 3. The original INTAC emissions are quantified in units of Mg/grid/month,
with a temporal resolution of one month and a spatial resolution of 0.1° x 0.1° for the year 2017. It is important to note that a
higher spatial resolution inventory, the 1-km resolution by MEIC is also available (Zheng et al., 2021). However, this 1-km
inventory only offers one data point per grid per year, while also providing insight into emissions from 2013, which in China
are quite different from those in 2019. For this reason, we have used the INTAC inventory herein since it more closely matches
the 2019 TROPOMI data. This dataset covers mainland China and includes emissions from eight sectors: power, industry,
residential, transportation, agriculture, solvent use, shipping, and open biomass burning (Wu et al., 2024). To align the
resolution of the original INTAC Inventory with that of TROPOMI grids, we undertake several processing steps: 1) The units
are converted from Mg/grid/month to $\mu g/m^2/s$ as the first step, due to the varying areas of each longitude-latitude grid. 2) Next,
the INTAC inventory is adjusted to a 0.05° x 0.05° grid using the nearest neighbor method. 3) Finally, we assume that the
monthly emissions remain constant on a day-to-day basis. To ensure that the values used do not fall within the error range of
the TROPOMI sensor (i.e. noise), values below 0.2 $\mu g/m^2/s$ are designated as NaN and are not considered further in this study.
**2.4 Wind Data**
The parameters of wind speed and direction are used from ERA-5 reanalysis (Hersbach et al., 2018, 2020). To correspond
with the overpass timing of TROPOMI, this study employs the average value of the u and v wind products, recorded hourly at
5:00 and 6:00 UTC. The specific product used is taken at a spatial resolution of 0.25°x0.25° to facilitate a more accurate
representation of the atmospheric column conditions (https://www.ecmwf.int/en/forecasts/dataset/ecmwf-reanalysis-v5), and
is subsequently interpolated onto the same TROPOMI 0.05°x0.05° grid in space and time. Since many of the areas considered
in this work are low laying urban conglomerates, with most of the terrain situated below an elevation of 500 m, wind data at
the 950 hPa level was selected.

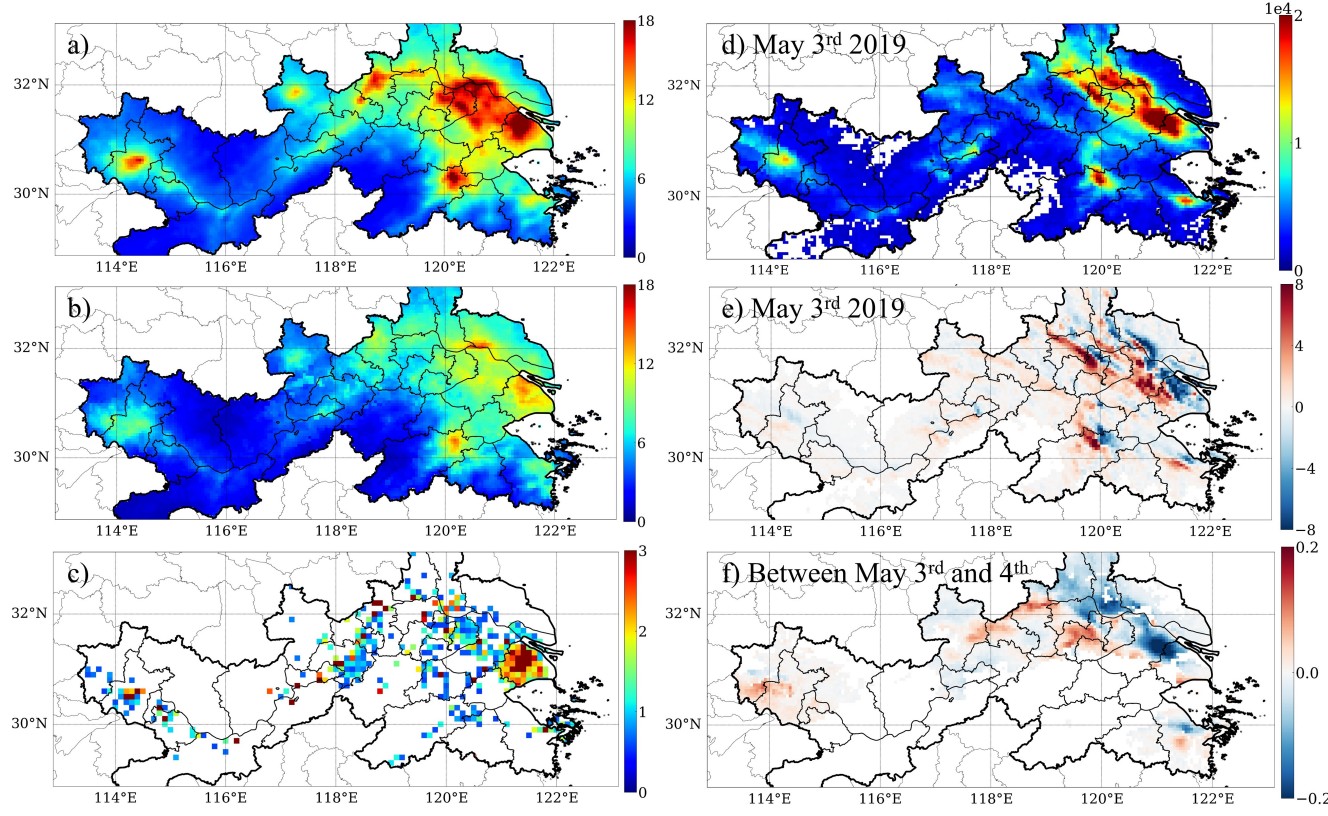

**Figure 2: (a) TROPOMI daily climatology of NO₂ column loading [10¹⁵ molec/cm²], (b) Standard deviation [STD] of daily NO₂ column loading [10¹⁵ molec/cm²], (c) INTAC monthly climatology of NOₓ emissions [μg/m²/s]. Data from May 3ʳᵈ 2019: (d) TROPOMI NO₂ column loading [μg/m²], and (e) gradient of wind multiplied by TROPOMI NO₂ column loading [μg/m²/s]. (f) The temporal derivative of TROPOMI NO₂ column loading between May 3ʳᵈ and May 4ᵗʰ [μg/m²/s].**

**2.5 Inverse model**

This study develops a flexible model based on first-order physics, chemistry, and thermodynamics and the continuity equation (mass conservation of trace species in the atmosphere) to approximate the emissions of NOₓ as shown in Figure 3 (Li et al., 2023a; Qin et al., 2023). Given a set of chemical substances in the atmosphere ($i = 1, \dots n$) with molar fractions (or mixing ratios) $C_i$, the vector $C = (C_1, \dots C_n)^T$, can be solved for based on conservation of mass following in a fixed Eulerian

reference system following Equation(1), where $v$ is the 2-D wind vector, $P_i$ and $L_i$ are the production and losses of $i$ (which
may include contributions from species), $E_i$ is emissions and $D_i$ is the sum of wet and dry deposition.

$$\frac{\partial C_i}{\partial t} = -\nabla \cdot (vC_i) + P_i(C) - L_i(C) + E_i - D_i \qquad (i = 1, \ldots n) \qquad (1)$$


The local rate of change of the column loading with time ($\partial C_i / \partial t$) is expressed as the sum of the input minus the output
of the transport (i.e., gradient transport $v \cdot \nabla C_i$ and pressure transport $C \cdot \nabla v_i$) and the net local output ($P_i(C) - L_i(C) + E_i -$
$D_i$). Note that in the case that the wind field is non-divergent, the gradient term $\nabla \cdot (vC_i)$ reduces to term $v\nabla C_i$ (Sun, 2022). In
this work, the chemical substances $C_i$ are generalized as TROPOMI NO$_2$ VCDs, which are denoted as $\Omega_{NO_2}$. The rate of change
of $\Omega_{NO_2}$ in the troposphere can be determined by a balance between emissions, chemical/physical losses, and transport of the
two individual terms NO and NO$_2$ by assuming that at the time of emissions they are related to each other by the ratio $NO_x =$
$\alpha_1 * NO_2$, and then retaining $\alpha_1$ as one of the terms to be flexibly solved for later in order to ensure that the model fits the
observations from TROPOMI and INTAC. According to equation (1), and approximating the chemical loss as first order with
a lifetime of $1/\alpha_2$ and the transport factors as linear with a distance of $1/\alpha_3$, the following mathematical model (2) can be
constructed, where the emissions of NO$_x$, denoted as $E_{NOx}$. The terms are then rearranged to solve for the emissions in equation

04    (3).

$$\frac{d(\alpha_1 * \Omega_{NO_2})}{dt} = E_{NOx} + \alpha_2 * (\alpha_1 * \Omega_{NO_2}) + \alpha_3 * \nabla((\alpha_1 * \Omega_{NO_2}) * v) \qquad (2)$$


$$E_{NOx} = \alpha_1 \frac{d(\Omega_{NO_2})}{dt} - \alpha_2\alpha_1 * \Omega_{NO_2} - \alpha_3\alpha_1 * \nabla(\Omega_{NO_2} * v) \qquad (3)$$


The daily TROPOMI NO$_2$ columns, monthly INTAC emissions, daily temporal derivative and spatial gradient computed
and utilized to fit the terms α$_1$, α$_2$, and α$_3$ in equation are shown in Figures 2c-2f.
The first term in the equation (3) symbolizes the influence of the rate of change in NO$_2$ columns on the estimation of NO$_x$
emissions, more simply put if the concentration is higher on the second day, then there must have been an emissions source
larger than all other factors in balance, and if the concentration is lower on the second day, then there must have been sinks
larger than the emissions source. The denoted as $\alpha_1$ illustrates the linear ratio of $NO_2$ to $NO_x$ and is a function of the
thermodynamics of combustion when the NO and $NO_2$ are first formed, as well as in-situ atmospheric thermodynamics and
rapid chemical adjustment after the combusted plume is lofted into the air. There is a basis for the use of $\alpha_1$ which varies in
space and time from both a chemical engineering perspective (Le Bris et al., 2007; Schwerdt, 2006), as well as from an
observational perspective (Karl et al., 2023a). The formation of thermal nitrogen oxides ($NO_x$) is a process characterized by
the reaction of atmospheric nitrogen ($N_2$) with atmospheric oxygen ($O_2$) under high-temperature conditions, and the $NO_x$-to-
$NO_2$ rapidly achieves a local pseudo-steady state equilibrium. The formation of $NO_2$ and nitric oxide (NO) is significantly
influenced by thermal conditions. The NO is preferentially formed at temperatures exceeding 1200°C, when the temperature
surpasses 1100°C, thermal $NO_x$ becomes the predominant contributor to overall $NO_x$ emissions, reaching a peak when the
temperature exceeds 1600°C. The secondary term $\alpha_2$ in the equation signifies the physical and chemical production and
destruction of $NO_x$, which is intrinsically associated with the chemical lifetime of $NO_x$. And the third term introduces the
concept of horizontal flux divergence, denoted by $\alpha_3$, representing the advective and pressure-induced atmospheric transport
of $NO_x$.
In this work, the divergence is computed using a second-order central difference method. The terms $\alpha_1$, $\alpha_2$, and $\alpha_3$ are fit
month-by-month and grid-by-grid (at 0.05°x0.05°) when and where data is available (including INTAC) using multiple least
squares regression. Certain extreme values of $\alpha_1$, $\alpha_2$, and $\alpha_3$ are mathematically computed, but are not physically plausible, and
in these cases are discarded from further consideration. Specifically, grids exhibiting a $NO_x/NO_2$ ratio less than 1, a positive
chemical loss term or chemical lifetime of $NO_x$ less than 30 minutes are designated as NaN. Subsequently, in each month and
on each grid, $\alpha_1$ is sampled over 10000 times within the 20th and 80th percentile of the computed probability density function
(PDF). For those grids which already have fitted values of $\alpha_1$, $\alpha_2$, and $\alpha_3$, in any given month, the bootstrap method is not
applied, and the fitted values are used for each day in that given grid. If either the grid does not have $\alpha_1$, $\alpha_2$, and $\alpha_3$, or it does
but not during the month being used, then the bootstrap method will still be used to compute the emissions and uncertainty
range.
On a daily and grid-by-grid basis where there is TROPOMI $NO_2$ column data and wind data, and the temporal derivative
and spatial gradient are computable, the following bootstrap method is used to compute the emission and the uncertainty range.
First, the distribution of $\alpha_1$ and corresponding $\alpha_2$ and $\alpha_3$ from the same month at all points in the region are resampled 1000
times per grid. Using the resampled coefficients, the model given in Equation 3 is finally used to compute the emissions of
$NO_x$ on a grid-by-grid and month-by-month basis. The mean of each grid-by-grid distribution of runs is hereafter assigned as
the mean emissions, while the standard deviation of each grid-by-grid distribution of runs is hereafter assigned as the range of
emissions uncertainty in that grid and on that day.

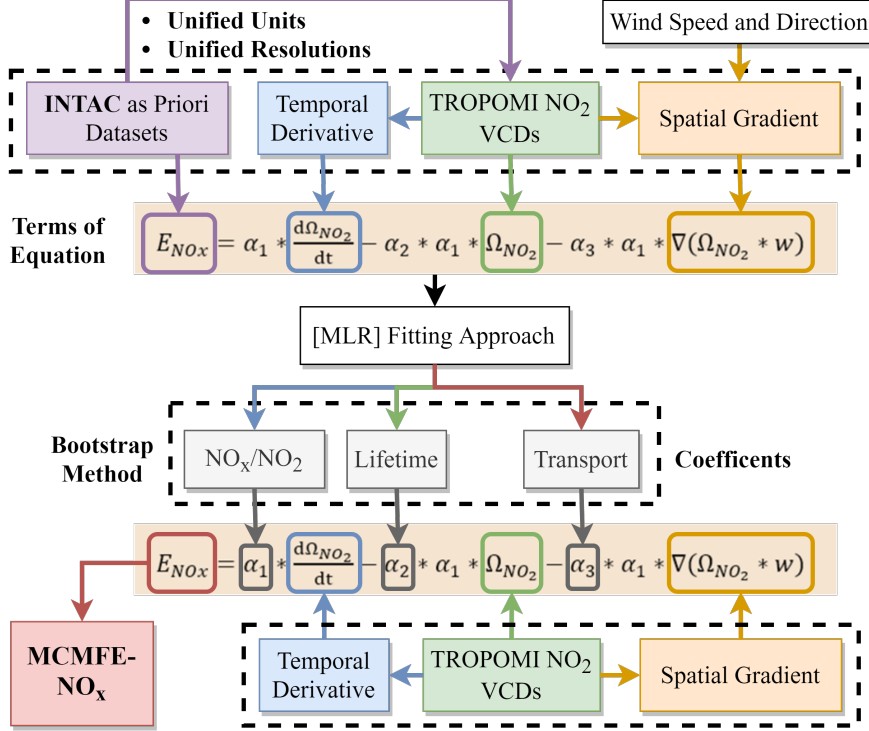

**Figure 3: The framework of the mass-conserving approach (MCMFE-$NO_x$).**

**44**    **2.6 Location of Sources**

**45**    An important objective of this study is to analyze the emission and thermodynamic characteristics of various emission

**46**    sources. To achieve this, the location data of five different high energy use facilities which operate under different power,

**47**    thermodynamic, and other conditions including power plants, steel and iron industries, heat production and supply, cement

**48**    factories, and biomass burning are selected as the input parameters for the distribution calculations. The location data of each

**49**    of these types is obtained from the Pollutant Discharge Permit Management Information Platform of the Ministry of Ecology

**50**    and Environment (http://permit.mee.gov.cn), which contains the information of these emission sources (name, city, latitude,

**51**    and longitude). It is important to note that not all these sources are of sufficient scale to be equipped with Continuous Emission

**52**    Monitoring Systems (CEMS) for emissions monitoring. Many of these sources are small to medium-sized industries, which

**53**    do emit pollutant gases such as $NO_x$ and have applied for formal discharge permits. The location data enables us to correlate

**54**    satellite observations with identified emission sources, thereby providing valuable insights into emission patterns and their

**55**    thermodynamic characteristics.

**56**    **3 Results and discussion**

**57**    **3.1 Coefficients results**

**58**    To examine the robustness of the coefficients results to the choice of study regions, the results obtained from the urban

**59**    areas Suzhou, Nanjing and Shanghai are resampled and refitted between the original fit's 20th and 80th percentiles, and the

**60**    results of the updated statistical distribution of monthly $\alpha_1$ values is compared with the original distribution. The resulting

**61**    distributions of $NO_x/NO_2$ ($\alpha_1$), lifetime (related to $\alpha_2$), and transport distances (related to $\alpha_3$) (Table 1) over the two cases are

**62**    nearly identical, demonstrating the stability of the MLR fitting method when used in connection with the emissions model, the

**63**    physical constrains employed on the fitted values, and sampling the 20th through 80th percentiles. Unlike existing models that

offer limited ranges (Beirle et al., 2019), this work accommodates higher variability and conforms to empirical observations
(Karl et al., 2023a; Laughner and Cohen, 2019). The annual percentiles from 20% to 80% for $\alpha_1$ values in regions 1, 2, and 3
are observed to be within the intervals of 3.9 to 19.0, 2.9 to 15.0, and 4.4 to 22.2 respectively, while the lifetimes respectively
from 0.34 to 0.60, 0.28 to 0.67, and 0.25 to 0.62 days. Overall, the community has assumed that negative transport, or net
export from highly emitting boxes, dominates the transport. In specific, it has generally been assumed that emissions exit from
an urban area, and the impact of upwind sources entering into the background of an urban area or source are frequently not
considered. However, the results herein show that this is actually only the case 55%, 49%, and 54% of the time over the three
domains respectively. This means that a significant amount of mass is transported into emitting areas from upwind emitting
areas, and is consistent with the computed positive transport (net import) values of 45%, 51%, and 46%. There are some
theoretical studies and case studies which have demonstrated that this is the case, but none have used observations over such
a long time period to analyze the frequency of occurrence (Cohen et al., 2011; Cohen and Prinn, 2011; Wang et al., 2023).
The sensitivity of the fitted coefficients ($\alpha_1$, $\alpha_2$, and $\alpha_3$) remain relatively stable to the changes in the a priori $NO_x$ emissions.
In specific, we design a perturbation run in which the emissions are randomly altered day-by-day and grid-by-grid from the
priori dataset near the extreme upper and lower bounds of their ±30% uncertainty range. This is then used in combination with
the original values from TROPOMI to refit the coefficients, as given in Table S1 and Figure S1. It is observed that over 60%
grids of the $NO_x/NO_2$ ratios and lifetimes, 40% grids in terms of the transports term are found to be robust, i.e., have a change
smaller than the 30% perturbed a priori emissions.
**Table 1: Ranges of $NO_x/NO_2$, lifetime and transport distances computed from annual dataset respectively at 20%, 50% and 80%**
**from region 1, region 2 and region 3.**

| Coefficients | Percentile | Region1 | Region2 | Region3 |
|---|---|---|---|---|
| | 20% | 3.5 | 2.8 | 4.2 |
| $NO_x/NO_2$ | 50% | 7.8 | 6.3 | 10.2 |
| | 80% | 17.1 | 14.6 | 21.4 |
| Lifetime (days) | 20% | 0.34 | 0.29 | 0.26 |

| | | | | |
|---|---|---|---|---|
| | 50% | 0.47 | 0.48 | 0.45 |
| | 80% | 0.58 | 0.69 | 0.63 |
| | 20% | -74.1 | -97.1 | -87.7 |
| **Transport/net export (km)** | 50% | -176. | -252. | -238. |
| | 80% | -392. | -548. | -522. |
| | 20% | 86.0 | 108. | 86.3 |
| **Transport/net import (km)** | 50% | 193. | 272. | 245. |
| | 80% | 432. | 550. | 533. |

The monthly distribution of $NO_x/NO_2$ ($\alpha_1$) and lifetime across various grids are presented in Figures 4a-4c. The parameter
$\alpha_1$ as observed across all three research areas demonstrates a peak in July and August, a minimum in December and January,
a second peak in February in the mean, $60^{th}$ percentile and $80^{th}$ percentile cases, and does not follow a standard seasonal pattern.
When looking at $\alpha_1$ on a region-by-region basis, the underlying factors become more clear. Region 1 and 2 exhibit a relatively
consistent $\alpha_1$ value with the overall pattern described above, with the only difference being region 1 has a secondary peak in
February while region 2 does not. In these cases, the pattern is closely related to both the atmospheric temperature and the
demand for excess power for heating as the centralized systems in the north of the country do not shut off until early March.
Region 3 records a markedly elevated monthly $\alpha_1$ compared to other regions from October through April, with a second overall
annual peak during March at the $80^{th}$ percentile and April at the $20^{th}$ percentile. This is again consistent with the atmospheric
temperature experienced in the Asian Monsoon region, and the extreme extra energy required for air conditioning during the
dry and hot times during February to April that occur annually, frequently rivaling those of the summer when it is more cloudy
and rains more. An important final finding is that the mean value of $\alpha_1$ is biased and always is found to be in between the $60^{th}$
and $80^{th}$ percentiles.
With respect to the lifetime of $NO_x$, the month-to-month value and variability of the mean and $60^{th}$ percentile are similar
to each other, while the variability of the 20th, 40th, and 80th percentiles are all larger. At the $20^{th}$ percentile, November and
December experience a longer lifetime than the rest of the year, consistent with reduced UV radiation. February deviates from
the other months, consistent with economic and energy demands as well as emissions overall being very different during the

the Chinese New Year period. In specific, the $80^{th}$ percentile lifetime has its longest annual value, while the $20^{th}$ and $40^{th}$ percentiles have their shortest annual values, indicating that high spatial and temporal variability exists with the emissions response to the movement of 500-800 million people over the annual 2-week long holiday. Similarly, the mean value of lifetime is found to be biased between the respective $60^{th}$ and $80^{th}$ percentile values.

Additionally, Figure 4d presents the mean $NO_x/NO_2$ values of various cities. The lowest values, consistent with few to no industrial sources and high levels of vehicle and residential use, are found respectively in Wuxi and Macau, both of which are known as high GDP and low energy-intensive production cities, and both of which are economically advanced. The next tier levels are observed in the well known urban areas like Beijing, Nanjing, Suzhou, and Hangzhou, which are similarly economically advanced and have high levels of car usage and public transportation, but also have some factories and industry. The next tier is found in places like Shanghai, Qingdao, Hong Kong, Nanjing, and Wuhan, which are similar to the tier above, but also combine significant sources related to shipping and industries co-related including refining and other more heavy industires. The highest values are found in Heze, Lianyungang, and Beihai, all of which have a large amount of heavy industry, coal and oil based industries for both energy and materials production, large ports, and other energy inefficient sources, as well as lower overall vehicle penetration rates and a rapidly growing economy. It is interesting to note that there are some exceptions, such as Maanshan, which is lower than expected, since it is economically similar to Heze, Lianuyungang, and Beihai, and has considerable coal industry. Moreover, this location also has a large amount of biomass burning to clear agricultural waste.

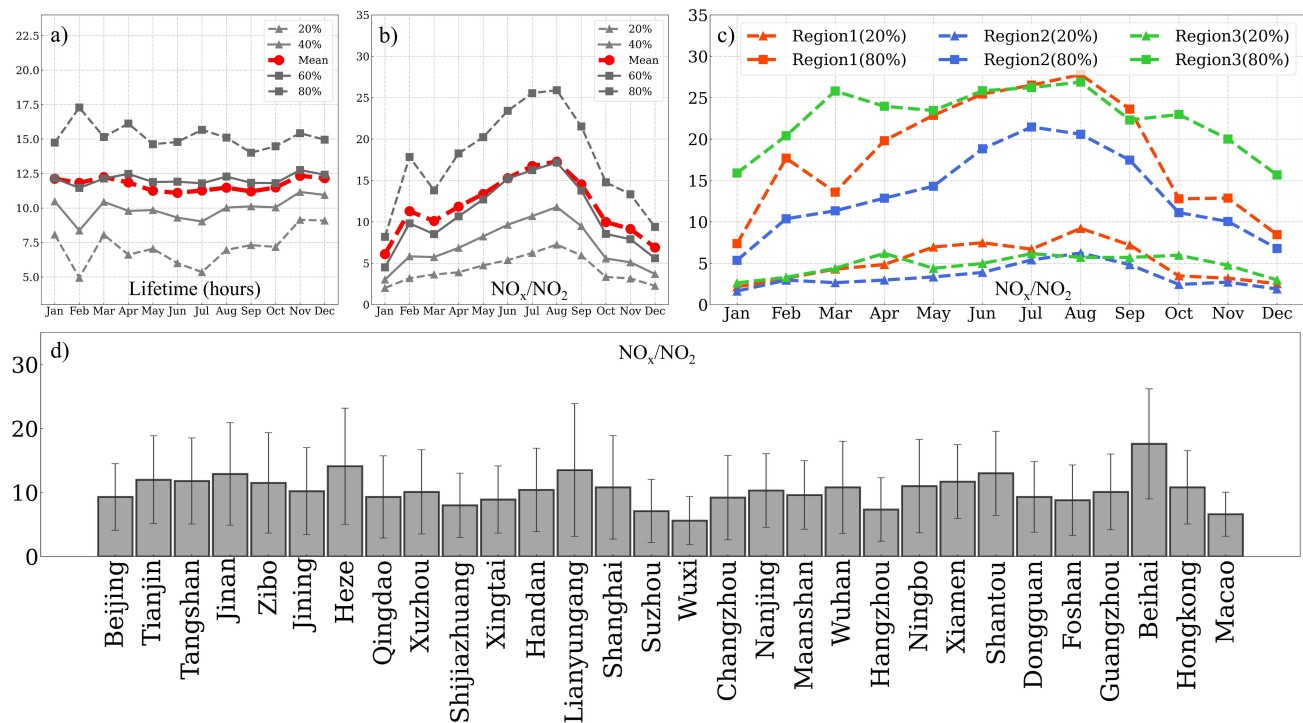

**Figure 4: The distribution (mean values, 20th, 40th, 60th, and 80th percentile values) of monthly (a) lifetime; (b) $NO_x/NO_2$; (c) 20th and 80th percentile values of monthly $NO_x/NO_2$ in three regions; (d) Mean values of $NO_x/NO_2$ over 30 cities**

This work analyzes the measure and distributions of $NO_x/NO_2$ over five different identified industrial source types: power plants, steel and iron factories, cement factories, heat production and supply, and biomass burning. The spatial distribution of five emission source types and their temporal median $NO_x/NO_2$ values in Region 1 are presented in Figure S2, with statistics of grids within different ranges of $NO_x/NO_2$ given in Table S2. The proportion of grids with $\alpha_1$ values exceeding 10 continues to exhibit a distinct difference between three groups: steel and iron factories (up to 52%), power plants (intermediate values, about 40%), and cement factories, heat production and supply, and biomass burning (lower values). Even though the emissions rapidly adjust from the hot air emitted at the stack or pipe exit, this is clearly significantly influenced by the thermodynamics of combustion itself, as well as additional factors including $NO_x$ control technologies (LNB and SCR), combustion 8 technologies (related to the heat rates and efficiency) and local policies. These results demonstrate clearly that the original

thermodynamic conditions still significantly influence the $NO_x/NO_2$ values at the scale observed by TROPOMI. The rationale
for this analysis is that each of these types of combustion sources has a very different set combustion temperatures, oxygen
availability, and other properties. Through both monthly distributions (Figures 5a-5e) and annual analysis of the PDFs of $\alpha_1$
values (Figures 5g-5k), it is clearly demonstrated that $\alpha_1$ has a significantly different set of characteristics across the different
sources.

33       Thermal power plants primarily focus on electricity generation, with the maximum operational temperature reaching up

to 2000°C, and the steel and iron factories utilizing blast furnaces operate at high flame temperatures ranging from 1350°C to
2000°C. As a result, the $NO_x/NO_2$ computed over the grids encompassing the two high-temperature sources is noticeably
higher when compared to other sources. The monthly values over steel and iron factories are slightly less stable (more variable)
than power plants. In the uppermost ranges (80th and 90th) of the PDF, the values corresponding to steel and iron factories
exceed those of power plants and other sources, likely due to the extremely high temperatures used in the production of certain
high-grade stainless steel. The manufacturing of cement involves combustion in a clinker at around 1000°C for preheating,
and in a precyclone tower at around 1400°C to complete the process of chemical reaction. Therefore, while in general the
values from cement factories are relatively high, but lower than the power plants and steel factories, and will favor $NO_2$ during
part of the process and NO during a different part of the process. As expected, it is found that the values of $\alpha_1$ for Cement are
lower than power plants and steel and iron factories, but higher than the other source types. Heat production and supply
generate steam and hot water through boilers and other devices, as well as export of heated water or steam. These are similar
to power plants but operate at a lower temperature and efficiency. Accordingly, this factor also has relatively low $\alpha_1$ values in
each month. However, July of the heat production and supply is an exception with high values as the hottest time of the year
when extreme numbers of people turn on the AC. The combustion of biomass includes uses for power, brick kilns, residential
use, as well as simply open biomass burning across the chains of agriculture, forestry, industrial waste, and municipal waste
as raw materials. The combustion can be done directly or after gasification, in both cases occuring with temperature lower than
1200°C, and possibly very low in the case of biomass burning. For these reasons, the $\alpha_1$ values of biomass burning are the
lowest of all the types. The temporal variations of the 80th percentile values for different industrial types exhibit distinct
temporal patterns (Figure 5f): Power plants, heat production and supply, and biomass burning have the highest values in July;
cement factories shows a bimodal distribution with peaks in July and August; steel and iron factories display a delayed response
with maxima in August and September.
The distributions of these sources exhibit substantial variability within and between their respective percentile ranges
(refer to Table 2). Since the $NO_x/NO_2$ values are derived exclusively from satellite observations and surface measurements,
any clear means of separating different underlying source types based solely on $\alpha_1$ will yield a way to attribute from space the
type of underlying emissions source. First, it is clearly observed that the 50$^{th}$ percentile range allows clear differentiation
between three groups: steel and iron factories (high value), power plants (medium value), and cement factories, heat production,
and biomass burning (low value). Although biomass burning is slightly lower than the other two in this group, the difference
is still smaller than between the three large groups. A second clear metric is formed when analyzing the ratio between the
difference of the 90$^{th}$ percentile and 80$^{th}$ percentile and the difference between the 20$^{th}$ percentile and 10$^{th}$ percentile (hereafter
called the high-low ratio or hl_ratio) following equation (4).

$$ \text{hl\_ratio} = \frac{(90\% - 80\%)}{(20\% - 10\%)} \quad (4) $$

The hl_ratio clearly differentiates between three groups: cement factories (high value), heat production and supply
(medium value), and power plants, steel and iron, and biomass burning (low value). Although biomass burning is slightly
lower than the other two in this group, the difference is still smaller than between the three larger groups. Merging the 50$^{th}$
percentile factor (high, medium, low) and the hl_ratio factor (high, medium, low) allows for unique attribution of the 5
underlying source types, following Table 2.

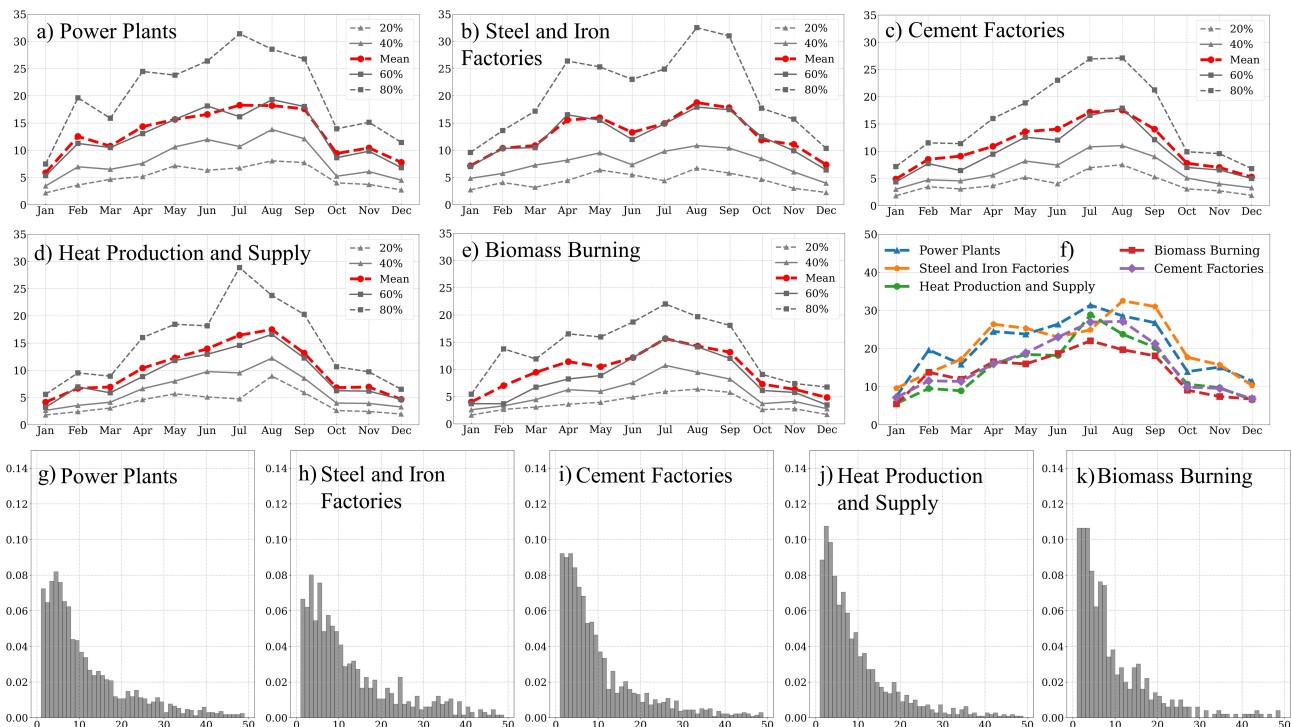

**Figure 5: The distribution (mean values, 20th, 40th, 60th, and 80th percentile values) of monthly NOx/NO₂ over grids from different sources (a) Power Plants; (b) Steel and Iron Factories; (c) Cement Factories (d) Heat Production and Supply; (e) Biomass Burning. (f) The distribution of 80th percentile values of each source; (g-k) Probability density functions (PDFs) of annual NOx/NO₂ of each source**

**Table 2: Ranges of NOx/NO₂ from five different industrial source types respectively at 10%, 20%, 50%, 80%, 90% and and high-low ratio (hl_ratio) hereafter defined as: (90%-80%)/(20%-10%). Attribution is achieved by the color fields: red (high, H), yellow (medium, M), blue (low, L), given in the right column in terms of the [50th percentile factor , hl_ratio factor].**

| Industrial Source Types | 10% | 20% | 50% | 80% | 90% | hl_ratio | Attribution Factors |
|---|---|---|---|---|---|---|---|
| Power Plants | 2.42 | 3.78 | 8.02 | 18.02 | 26.43 | 6.15 | [M,L] |
| Steel and Iron Factories | 2.41 | 3.87 | 9.21 | 21.38 | 30.38 | 6.16 | [H,L] |
| Cement Factories | 2.16 | 3.14 | 7.14 | 16.50 | 25.27 | 8.93 | [L,H] |
| Heat Production and Supply | 2.15 | 3.03 | 6.93 | 15.03 | 21.96 | 7.94 | [L,M] |
| Biomass Burning | 1.93 | 2.92 | 6.50 | 15.15 | 21.05 | 5.96 | [L,L] |

## 3.2 Emission results

The annual mean and standard deviation of the daily emissions and the annual mean of the daily uncertainties are given in Figures 6a-6c with the day-to-day results available for download at https://figshare.com/s/38731b16156be53a7c0b (for review purposes only, will be converted into a permanent doi upon acceptance). The daily average emissions and uncertainties of these selected representative urban areas are computed as follows: In region 1, Beijing, Tianjin, and Tangshan which are primarily coal-based and oil-based resource areas have values of $1.6\pm0.8$ µg/m$^2$/s, $2.3\pm1.0$ µg/m$^2$/s, and $2.4\pm1.1$ µg/m$^2$/s, respectively. Jinan and Zibo, rapidly industrializing locations, have emissions of $1.7\pm0.9$ µg/m$^2$/s and $1.8\pm0.8$ µg/m$^2$/s. In region 2, $NO_x$ emissions in Shanghai are high at $2.0\pm0.5$ µg/m$^2$/s. Cities like Nanjing, Suzhou, and Wuhan, which have experienced rapid economic development, show values of $1.4\pm0.6$ µg/m$^2$/s, $1.5\pm0.7$ µg/m$^2$/s, and $1.2\pm0.5$ µg/m$^2$/s, respectively. Ma'anshan, with a rapidly developing industry, also has high emissions at $1.2\pm0.5$ µg/m$^2$/s. In region 3, cities near the Pearl River estuary engaged in wharf ship movement, such as Hong Kong, has emissions of $1.8\pm0.7$ µg/m$^2$/s. Cities like Dongguan and Foshan, which have undergone significant industrialization, show emissions of $1.7\pm0.6$ µg/m$^2$/s and $1.7\pm0.5$ µg/m$^2$/s. The 25[th] percentile, mean, and 75[th] percentile values of daily and grid-based emissions (T/day) for 30 cities across three regions (as listed in Figure 1) are detailed in Table S3.

A comprehensive sensitivity analysis has been conducted to assess the robustness of MCMFE-$NO_x$ (Lu et al., 2024). The degrees of freedom of the framework are detailed in the supplementary materials, which provide a robust justification for the daily estimation approach. A set of uncertainty simulations is uniformly applied as the $TO_{40\%}$ case, where the $NO_2$ columns are multiplied by random perturbations ranging from 0.6 to 1.4. By accounting for the buffering effects of the chemical and thermodynamic terms, our findings demonstrate that the mass-conserving flexible emissions inversion method yields robust inversion results (as presented in Figure S3) when compared to the traditional wind speed and concentration gradient method. It is observed that 93% of the daily grid cells exhibited a ratio $[(TO_{0\%}-TO_{40\%})/TO_{40\%}]$ within $\pm40\%$. The day-by-day and grid-

by-grid NO$_x$ emission ranges are quite similar in both cases (as presented in Figure S4). These findings indicate that changes
in the driving factors ($\alpha_1$, $\alpha_2$ and $\alpha_3$) across different NO$_2$ column loading scenarios are generally smooth and consistent.

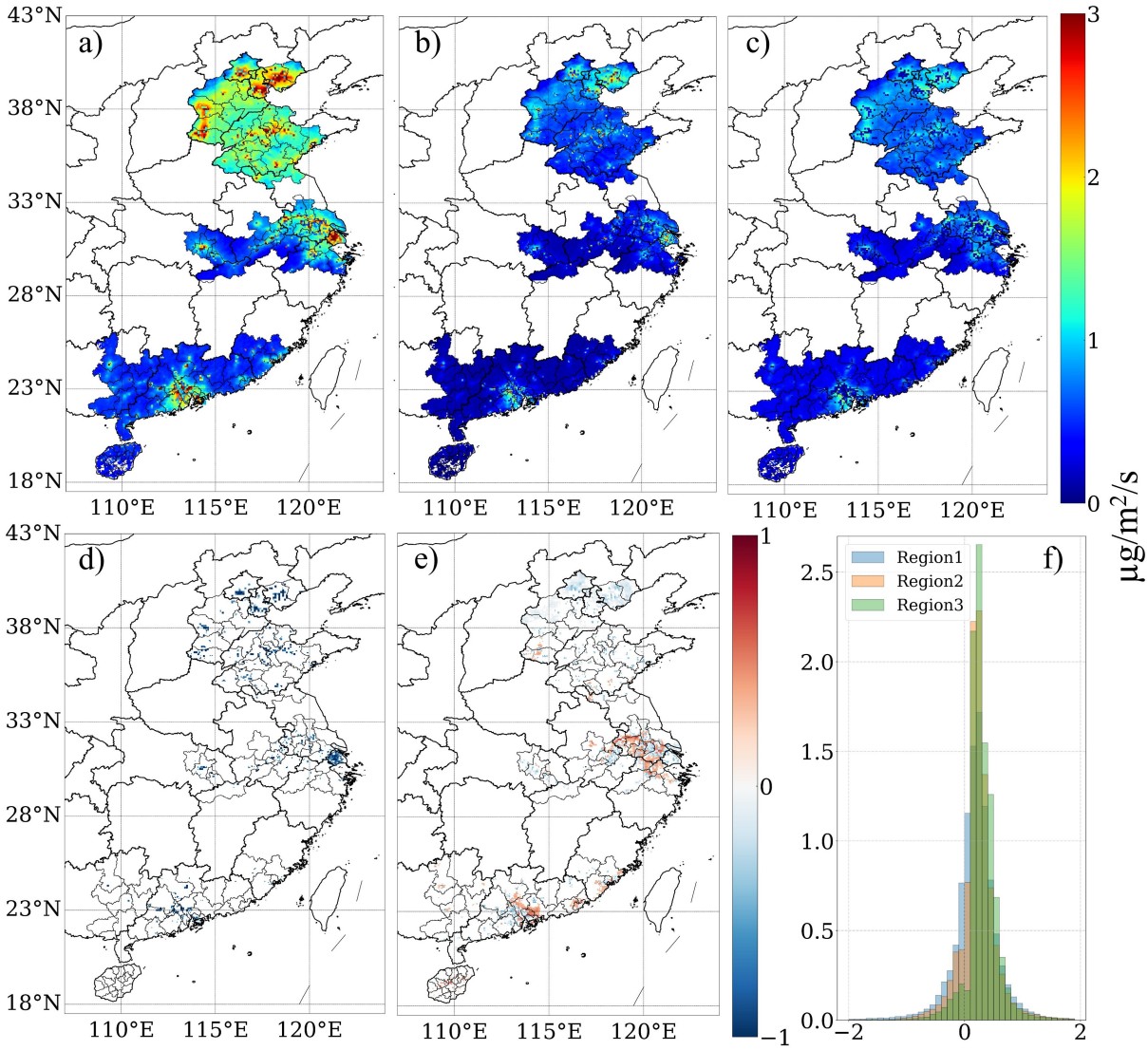


**Figure 6: Representations of daily computed MCMFE-NO$_x$ [μg/m²/s]: (a) climatological mean of day-to-day emission, (b)**
**climatological standard deviation of day-to-day emission, (c) climatological mean of day-to-day uncertainty. The differences between**
**uncertainty and standard deviation [μg/m²/s] of: (d) the locations where the uncertainty is smaller than the standard deviation (Diff**
**< -0.5), (e) the locations where the uncertainty is similar to or larger than the standard deviation (-0.5 < Diff < 0 & Diff > 0.3), (f) the**
**PDF of monthly differences in three regions.**
There is minimal overlap between regions with high day-to-day variability and regions with high uncertainty. In Wuhan,
for example, high variability is observed in the city center, while high uncertainty is located north of the city near the river
area. Regions shown in Figure 6e where uncertainty is similar to (less than 0.5 $\mu g/m^2/s$ lower) or exceeding 0.3 $\mu g/m^2/s$ higher
than day-to-day variability are undergoing land-use changes, indicating that more robust validation and retrieval algorithms
may be required in these regions. The forms of land use and land cover changes, such as urbanization deforestation, agricultural
expansion, and infrastructure development, can significantly impact $NO_x$ emissions through various mechanisms. These
regions encompass the southern part of Hebei province, industrializing locations in Shandong province, suburban areas around
Xuzhou, Suzhou, Wuxi, Changzhou, Zhenjiang, and Nanjing in Jiangsu province, the northern expanded part of Wuhan,
developing cities in Guangdong province and Xiamen. They are situated in suburban or rapidly developing rural areas that
were previously overlooked by the a priori datasets, covering 22%, 24%, and 12% of region 1, 2 and 3, respectively. In contrast,
Figure 6d illustrates that many metropolitan areas such as centers of Beijing, Tianjin, Shanghai, Hong Kong, Guangzhou,
Suzhou, Changzhou, Nanjing, Hangzhou, Wuhan, and Xuzhou where land surfaces are not changing significantly, exhibit over
0.5 $\mu g/m^2/s$ smaller uncertainty than day-to-day variability. These grids cover approximately 6%, 5%, and 2% in region 1, 2
and 3, respectively. This study highlights the importance of considering day-to-day variability in emission calculations for
these areas, emphasizing the limitations of relying on monthly or annual averages from a small sample of daily data.
Additionally, this research includes a comparison of the monthly mean of uncertainty and monthly variability of emission,
illustrated in Figure 6f.
**3.3 Emission see-saw**
The differences between MCMFE-$NO_x$ and INTAC are outlined in Figure 7. Analysis of the daily differences across three
regions (Figure 7c) reveals that INTAC exceeds MCMFE-$NO_x$ on approximately 6.9%, 11.1%, and 8.4% of the grids in regions
1, 2, and 3, respectively. These grids cover small areas of the spatial domain and are located in the highly developed commercial
centers and sites with significant pollution, exhibiting emissions patterns consistent with enhancing energy efficiency,
successful abatement, or mitigation of $NO_x$ sources, and/or potential shutdowns (Figure 7b). However, INTAC tends to
underestimate values in more grids, particularly in region 1. This includes grid areas where the day-to-day discrepancies exceed
1 $\mu g/m^2/s$, indicating substantial sources that the a priori emissions have overlooked. The grids where the differences surpass
1 $\mu g/m^2/s$ constitute about 55%, 15%, and 7% in regions 1, 2, and 3, respectively. As evidenced in the climatological mean of
the differences, a considerable quantity of emission sources has been detected in suburban regions and swiftly evolving rural
areas, which are absent in the a priori datasets. The coverage of these grids in the region 1 is much larger than those from the
other two regions. The regions of Beijing, Tianjin, and Tangshan, as well as Jinan and Zibo in Shandong province, along with
Shijiazhuang, Xing Tai, and Handan in Hebei province, have experienced substantial growth and have been extensively
explored, with more active and new emission sites misidentified.  In region 2, the northern part of Wuhan city and the land
over the Yangtze River in Jiangsu Province, especially near Suzhou and Wuxi, exhibit higher emissions than those reported in
the a priori emissions. The urban core of Wuhan has remained stable over a long time due to its compact and developed nature
more than two decades ago, but the outward expansion towards the northern sectors is new and not well constrained by the a
priori data. Over the Yangtze River, some of ignored emissions are not accounted for in the INTAC dataset. A portion of these
emissions is attributable to development along the river, such as power plants, steel and iron plants located right next to the
river. Furthermore, certain areas within region 3 contain sources that are not updated in the a priori datasets. The grids located
on the southern periphery of Hong Kong are near the airport and wharf. Guangzhou has been focusing on the development of
extensive scientific zones in the eastern sector and is fostering growth in Nansha in the southern sector as a new district. Along
the boundary of Shenzhen, Dongguan is attracting industry from Shenzhen. This trend of new cities offering incentives is also
evident in Jiangmen, with individuals migrating from Guangzhou and Foshan and relocating to Jiangmen across the border.
Therefore, the higher values from MCMFE-NO$_x$ are in line with the actual local development situation and policies, which are
reasonable.

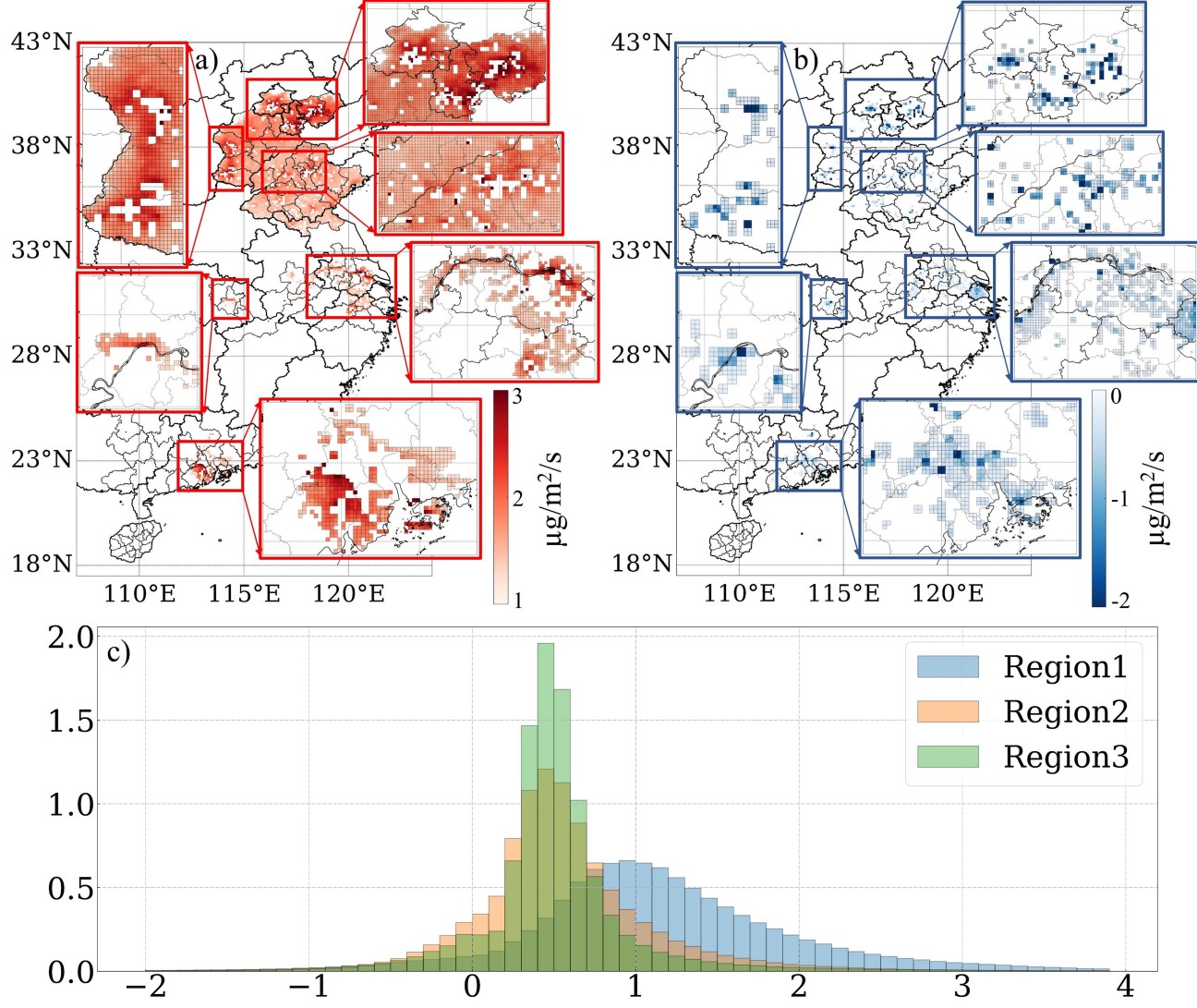


**Figure 7: Map of all grids which have at least 30 days during which the difference between MCMFE-NO$_x$ and INTAC is larger than 1.0 ug/m$^2$/s and smaller than 0 ug/m$^2$/s. (a) Climatological day-by-day mean only on those days which meet the difference being larger than the 1.0 ug/m$^2$/s cutoff. (b) Climatological day-by-day mean only on those days which meet the difference being smaller than the 0 ug/m$^2$/s cutoff. (c) PDF of all day-by-day and grid-by-grid differences on those grids which meet the cutoff, including those days which do not meet the cutoff over: Region1 (blue), Region2 (orange), and Region3 (green).**

## 3.4 Emissions over rivers

Emissions on and adjacent to rivers is an important research objective, as they are influenced by various aspects of anthropogenic activities, and require different surface data to retrieve $NO_2$ column information. There have even been previous studies reporting that the water itself or shipping activity on water may be the main source of $NO_x$ emissions (Kong et al., 2023; Zhang et al., 2023). The study regions of this paper includes the Yangtze River in Jiangsu section, Anhui section and Wuhan section, and the Yellow River in Shandong section, where the width of these rivers is close to or more than 5 km, allowing a pixel or more of retrieved $NO_2$ which is mostly or solely dependent on the river environment. Besides, there are numbers of emission sources burning coal like power plants and steel factories which are located right next to the river to pull the water for their cooling requirements, especially in Jiangsu section. The Figure 8a shows the locations of five power and industrial sources, including power plants, steel and iron factories, heat production and supply, CHP and cement factories along the Yangtze River. The total emissions in different sections are shown in the Table 3. The spatial distribution of total emissions (MCMFE-$NO_x$ and INTAC-$NO_x$) in the different sections of the Yangtze River and are demonstrated in Figures 8b-8d. The grids which contain these sources are highlighted with red frames, the annual total emission and uncertainty of these girds are also shown in Table 3.

Emissions and uncertainties over the Yangtze River in Jiangsu section, Anhui section and Wuhan section are 237±114 Kt, 157.67±54.86 Kt and 63.66±18.04 Kt respectively for the entire year, with high values in Nanjing, Yangzhou, Ma'anshan, Wuhu, Tongling and Wuhan sections. Correspondingly, emissions of INTAC are approximately 100 Kt, 98 Kt and 47 Kt. Over the Yangtze River in Jiangsu section, these highlighted grids account for 27% in number but contribute nearly 37% for emissions. And in Anhui section and Wuhan section, they represent 16% and 14% of the total number and contribution of 26% and 19% for emissions respectively. The MCMFE-$NO_x$ values for all grids, as well as the proportion of MCMFE-$NO_x$ from power and industrial sources, are both higher than those in INTAC. This discrepancy indicates a potential underestimation of

emissions from small and medium sources. Along the edges of the rivers, a vast amount of minor economic activities such as
utilization of machinery in agricultural practices, energy transfer devices employed in maritime activities, residential usage of
controlled combustion, and small-scale industrial enterprises are under reported. In the shipping sector of INTAC, $NO_x$
emissions account for less than 10% of the total emissions across all sectors, which is significantly lower than the estimates
derived from the total ship emission based on automatic identification system combined with China Classification Society
(Zhang et al., 2023). This study reports values as high as 83.5 Kt $year^{-1}$, which is 10 times greater than the shipping sector
estimates from INTAC. It also reveals that the MCMFE-$NO_x$, which includes contribution from all the sectors are unlikely to
be overestimated.
Besides, the Yangtze River bridges have significant impacts on the transportation system of these regions. Figure 8 also
shows the locations of the other Yangtze River bridges, which are also highlighted with yellow frames. The Wuhan Yangtze
River Bridge and the Nanjing Yangtze River Bridge, which were completed and opened to traffic earliest, are also dual-purpose
bridges for railways and highways. The emissions over the Wuhu Yangtze River bridge are the highest among them, and the
$NO_x$ emissions over the Nanjing Yangtze River bridge in INTAC are considerably lower than those in MCMFE-$NO_x$,
suggesting that heavy transport emissions in these specific grids are also underestimated. Table 3 presents the results of three
Yangtze River Bridges, including the Wuhu, Wuhan and Nanjing Yangtze River bridges.
Generally, the emissions over the Yellow River are lower than the Yangtze River, which aligns with expectations, as
presented in the Table 3. This can be attributed to the heightened caution exercised due to the river's lower water levels.
Consequently, there is no coal transportation along the river. Cities situated along the Yellow River, such as Zibo, possess
their own oil reserves. Chemical plants in these cities, which utilize oil as an energy source, operate at lower temperatures
compared to coal based power plants. Furthermore, it has been observed that emission values are elevated in the city center of
Jinan.

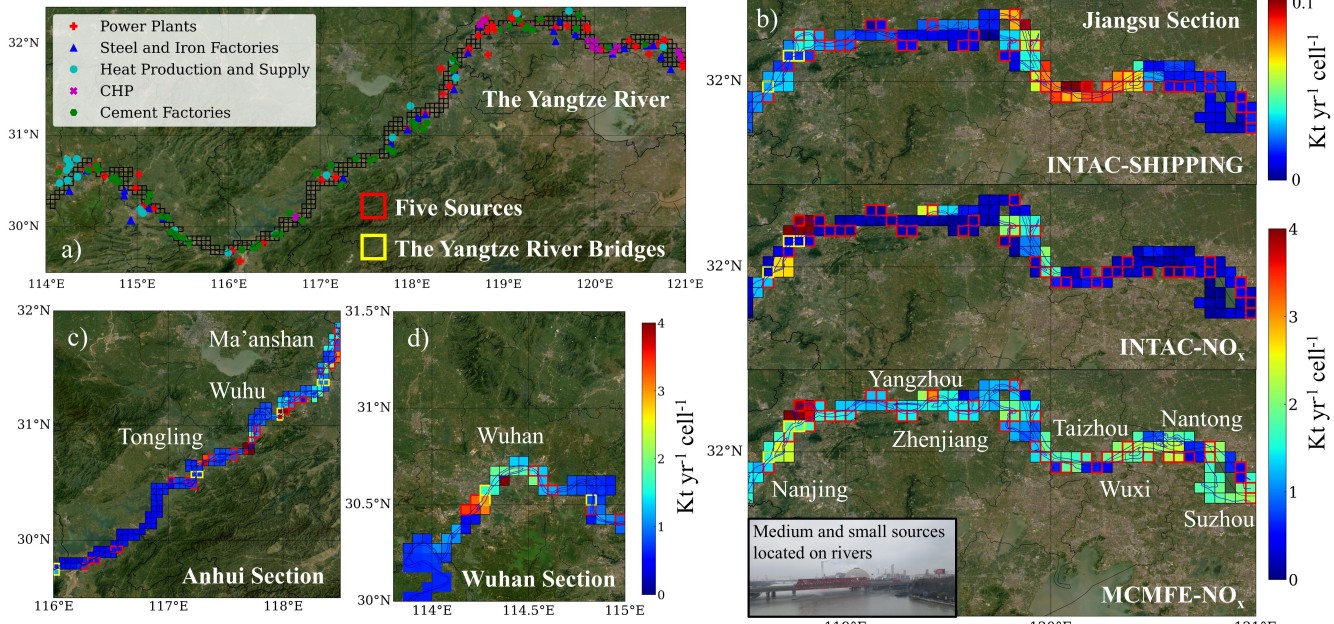

Figure 8: (a) The location of different sources along the Yangtze River. The total emissions (Kt yr$^{-1}$ cell$^{-1}$) over the Yangtze River in the (b) Jiangsu section: MCMFE-NO$_x$, INTAC-NO$_x$ (all sectors) and INTAC-NO$_x$ (shipping sector) (c) Anhui section: MCMFE-NO$_x$, (d) Wuhan section: MCMFE-NO$_x$. The grid cells are defined by a latitude-longitude grid with a resolution of 0.05° x 0.05°, meaning that area of each cell varies with latitude. This variation is accounted for in emission calculations to ensure accurate representation of emissions per unit area.

Table 3: Annual total NO$_x$ emissions and uncertainties (Kt year$^{-1}$) over the Yangtze River (MCMFE-NO$_x$, INTAC-NO$_x$ and the total ship emissions) and the Yellow River in different sections, power and industrial sources (the proportion of number and emissions) and Yangtze River Bridges.

| The Yangtze River | Jiangsu Section | Anhui Section | Wuhan Section |
|---|---|---|---|
| MCMFE-NO$_x$ | 237.±114. | 158.±54.9 | 63.7±18.0 |
| INTAC-NO$_x$ (All Sectors) | 100. | 97.8 | 46.6 |
| INTAC-NO$_x$ (Shipping Sector) | 7.63 | 8 | 3.4 |
| The Total Ship Emissions | 83.5 | | |
| Grids over Power and Industrial Sources | Jiangsu Section | Anhui Section | Wuhan Section |
| Proportion (Number of Grids) | 27% | 16% | 14% |

| | | | |
|---|---|---|---|
| MCMFE-NO$_x$ | 87.4±43.4 | 40.5±14.3 | 12.4±4.07 |
| INTAC-NO$_x$ (All Sectors) | 21.5 | 23.0 | 7.27 |
| Emissions Proportion (MCMFE-NO$_x$) | 36.8% | 25.7% | 19.5% |
| Emissions Proportion (INTAC-NO$_x$) | 21.5% | 23.5% | 15.6% |
| The Yangtze River Bridges | The Nanjing Yangtze River Bridge | The Wuhu Yangtze River Bridge | The Wuhan Yangtze River Bridge |
| MCMFE-NO$_x$ | 5.31±3.29 | 7.27±1.70 | 4.62±0.53 |
| INTAC-NO$_x$ | 0.93 | 6.60 | 4.99 |
| The Yellow River | Shandong Section | | |
| MCMFE-NO$_x$ | 158.±72.8 | | |

## 4 Conclusions

This work applies a model-free analytical approach that assimilates daily-scale remotely sensed tropospheric columns of NO$_2$ from TROPOMI in a mass-conserving manner to invert daily NO$_x$ emissions and the optimized underlying ranges of the driving chemistry, transport and physics. The results herein are presented over three rapidly changing regions in China, each located in different climatological zones. These regions encompass densely urbanized sub-regions, as well as surrounding rural, rapidly developing suburban and urbanizing sub-regions. Unlike traditional approaches that mainly concentrate on the Yangtze River Delta, Beijing-Tianjin-Hebei, and the Pearl River Delta, this research adopts a more comprehensive and uniform selection based on observations and climate zones. Notably, this work includes previously large cities such as Wuhan along the middle Yangtze River, Qingdao, Jinan, and others in Shandong Province, and Hong Kong, Shantou, and Xiamen along the South China Sea.

One important conclusion relates to the parameter $\alpha_1$, observed in three research areas, peaks in July or August and reaches a minimum in December and January due to UV radiation. Furthermore, $\alpha_1$ shows a second peak in February, reflecting varied economic and energy demands during the Chinese New Year period. Among the cities in research areas, the highest values are

found in Heze, Lianyungang, and Beihai, all of which have a large amount of industries. Source attribution is also quantified
with respect to the local thermodynamics of the combustion temperature, revealing distinct characteristics of $\alpha_1$ across five
industrial sources. The 50th percentile range and the hl_ratio allow clear differentiation and unique attribution of the five source
types. Note that the $\alpha_1$ values used herein are found to match well with observations in urban areas (Karl et al., 2023b) and
areas with large industrial sources (Li et al., 2023a; Lu et al., 2015), although they are far outside of the bounds currently used
by most models, indicating that the current generation of atmospheric models may not be able to capture such observed
emissions sources well (Beirle et al., 2019).
Several additional scientific points of interest are revealed regarding the MCMFE-NO$_x$ results. First, the day-to-day and
grid-by-grid emissions and uncertainties are found to be 1.96±0.27 µg/m$^2$/s on pixels with available priori values (1.94
µg/m$^2$/s), while 1.22±0.63 µg/m$^2$/s extra emissions are found on pixels in which the INTAC Inventory is lower than 0.3
µg/m$^2$/s. Some grids show lower MCMFE-NO$_x$ compared to INTAC, mainly in urbanized and polluted areas, possibly due to
energy efficiency, abatement efforts, or mitigation of NO$_x$ sources, and/or potential shutdowns. The illustration also highlights
the grid areas where the daily differences exceed 1 µg/m²/s, indicating significant sources missed by the priori datasets.
Second, rivers are a crucial research focus because they impact numerous aspects of human activities. Emissions of
industrial sources from missing sites adjacent Yangtze River are found to be 161.±68.9 Kt/yr, which is 163% higher than the
a priori. There are numbers of emission sources burning coal like power plants and steel factories which are located right next
to the river to pull the water for their cooling requirements, especially in Jiangsu province. Over the Yangtze River in Jiangsu
province, these highlighted grids account for 27% in number but contribute nearly 40% for emissions. And in Anhui province
and Wuhan city, they represent 16% and 14% of the total number with contribution of 26% and 19% for emissions respectively.
This set of findings indicate that the contribution from small scale industries in pixels on or adjacent to rivers offer a significant
source of unaccounted for $NO_x$ emission, which is shown to be larger than the amounts reported from biological sources on
lakes (Kong et al., 2023) and inland shipping activities (Zhang et al., 2023).
Third, there is little overlap between high day-to-day variability and high uncertainty. The uncertainty over land surfaces
which are not changing is smaller than the day-to-day variability, emphasizing the importance of considering day-to-day
variability in emissions. Conversely, uncertainty over areas experiencing land-use changes or over water is similar to or larger
than the day-to-day variability, indicating that more robust validation and retrieval algorithms may be required in these regions.
**Data availability**
All underlying data herein are available for access by the editors and reviewers at
https://figshare.com/s/38731b16156be53a7c0b and will be made available to the community upon publication. The TROPOMI
data used in this work is available for download at: https://data-portal.s5p-pal.com/products/no2.html. ECMWF wind speed
and direction are available for download at: https://cds.climate.copernicus.eu/cdsapp#!/dataset/reanalysis-era5-pressure-
levels?tab=form. The location data of industrial sources is obtained form the Pollutant Discharge Permit Management
Information Platform of the Ministry of Ecology and Environment (http://permit.mee.gov.cn).
**Author contributions**
This work was conceptualized by Jason Blake Cohen and Lingxiao Lu. The methods were developed by Jason Blake
Cohen and Kai Qin. Xiaolu Li and Qin He provided insights on methodology. Investigation was done by Lingxiao Lu, Kai
Qin and Jason Blake Cohen. Visualizations were made by Lingxiao Lu and Jason Blake Cohen. Writing of the original draft
was done by Lingxiao Lu and Jason Blake Cohen. Writing at the review and editing stages were done by Lingxiao Lu and
Jason Blake Cohen
**Competing interests**
The authors declare that they have no conflict of interest.
**Acknowledgments**
This study was funded by the National Nature Science Foundation of China (42075147, 42375125).

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
