# Peer review of "Identifying Missing Sources and Reducing NO$_x$ Emissions Uncertainty over China using Daily Satellite Data and a Mass-Conserving Method"

_EGUsphere, 2024_

## Author Response (AR1)

Dear Editors and Reviewers, we are providing a point-by-point reply to the reviewer comments below. In each case, all reviewer comments are copied and pasted below unedited in normal text, while our responses are given in blue text. We have worked diligently to respond to each and every comment, with corresponding changes in manuscript shown. We have computed new sensitivity runs, and produced multiple new figures and tables. In the end, the comments and reviews have helped us to produce an even stronger result, one which is clearly new, unique, and consistent. Thanks again to everyone for your patience, insights, and support. Looking forward to continued interaction and the production of an excellent and useful scientific product, with real world applications and demand.

Regards,

Jason Blake Cohen

jasonbc@alum.mit.edu

on behalf of the authors

Reviewer #1: This study uses an emerging mass-balance based analytical framework to estimate spatially resolved (daily) $NO_x$ emission fluxes and (monthly) associated properties of $NO_x$ sources and processes from high-resolution satellite (TROPOMI) retrievals in eastern China. The results are then interpreted to derive 1) insightful discussions regarding these properties, especially the $NO_x/NO_2$ emission ratio above different types of sources; 2) significant overestimation of $NO_x$ emissions above strong $NO_x$ sources by the a priori emissions inventory (MEIC), and underestimation in the downwind areas; and 3) strong industrial and shipping $NO_x$ above and adjacent to rivers in China. Most of the results and insights over the highly emitted study region are unprecedented in existing literature. The paper is overall well-written with sound methods and results. At the same time, some critical details are missing and need clarification. I support the publication of this manuscript, provided that the following comments can be addressed.

Main comments:

1) In the current description of the approach and Figure 3, it is unclear to me how the $NO_x$ emissions are assumed in the first fitting step to derive three monthly coefficients. Of course, they cannot vary everyday as in the second step (otherwise the

equations cannot be solved). Did you directly use MEIC emissions? Or did you assume monthly invariant emissions? Please clarify this important issue, and verify that your approach is robust against your assumption (the fitted alpha1-alpha3 changes little with imperfect assumptions on $NO_x$ emissions in the first step).

Thank you for encouraging us to provide clarification regarding the underlying processes used in our approach. We have directly used MEIC emissions. These emissions are given on a month-by-month basis in their original form. We have continued to use this same set of assumptions by assuming that each day's value is identical. To address your concern about the robustness of our approach, we conducted sensitivity analyses to verify that the fitted coefficients (alpha1-alpha3) are relatively stable even with imperfect assumptions surrounding the a priori $NO_x$ emissions used in the first step. As detailed uncertainty analysis shows, changing these values of the a priori emissions does not lead to a significant change in the fitted values of alpha1-alpha3, this is because there are physical constrains placed on these values, to ensure that they are reasonable. The specific analysis process is as follows:

The priori emissions dataset (INTAC) utilized in this study was developed by the MEIC team in collaboration with various authoritative research institutions. We consider it the most suitable choice for priori datasets over China. The uncertainties in INTAC arise from its integration process. Relatively low uncertainties are exhibited by $NO_x$ emissions of INTAC, benefiting from well-established estimates for large-scale combustion sources. Given this, we assume that the uncertainty is less than 30%. In this study, we selected Region 2 and tested what impacts would occur if the prior emissions were to have uncertainties near the extreme upper and lower bounds of their ±30% uncertainty range. The coefficients were refit using new priori emissions values ($INTAC_{30\%}$ Case) while maintaining the same values from TROPOMI. The results are presented in Response Figure 1. It is observed that for most grids (over 60%), the $NO_x/NO_2$ ratios and lifetimes exhibit significant robustness following the 30% perturbation of INTAC as the priori. However, the transport parameter demonstrates less robustness, with approximately 40% of grids showing a lower ratio than the 30% perturbation, meaning slightly over half of the total grids may suffer from a non-linear response. As illustrated in Response Table 1, the 20th and 80th percentile ranges of $NO_x/NO_2$ are from 2.8 to 14.6 and from 2.7 to 15.0 in the $INTAC_{0\%}$ and $INTAC_{30\%}$ cases, respectively. The corresponding ranges for lifetime (days) are from 0.29 to 0.68 and from 0.27 to 0.73. The differences in transport (km) are slightly larger than those of the other coefficients, ranging from -323. to 351. and from -289. to 314. Our findings indicate that the variations in the fitted coefficients are relatively minor, thereby confirming the robustness of our method. And any slightly larger changes in the transport term are buffered by the non-transport terms.

[Figure]

Response Figure 1. The distributions of three key coefficients obtained from INTAC$_{0\%}$ and INTAC$_{30\%}$: a) NO$_x$/NO$_2$, b) Lifetime [hours], c) Transport [km]; The ratio of (INTAC$_{30\%}$-INTAC$_{0\%}$)/INTAC$_{0\%}$ is also displayed on: d) NO$_x$/NO$_2$, e) Lifetime [hours], f) Transport [km];

Response Table 1. The 20$^{th}$, 50$^{th}$,80$^{th}$ percentile ranges of three coefficients from INTAC$_{0\%}$ and INTAC$_{30\%}$.

| NO$_x$/NO$_2$ | INTAC$_{0\%}$ | INTAC$_{30\%}$ |
|---|---|---|
| 20% | 2.76 | 2.68 |
| 50% | 6.30 | 6.24 |
| 80% | 14.6 | 14.9 |
| Lifetime (days) | INTAC$_{0\%}$ | INTAC$_{30\%}$ |
| 20% | 0.29 | 0.27 |
| 50% | 0.48 | 0.48 |
| 80% | 0.68 | 0.73 |
| Transport (km) | INTAC$_{0\%}$ | INTAC$_{30\%}$ |
| 20% | -323. | -289. |
| 50% | 18.1 | 16.3 |
| 80% | 351. | 314. |

In order to communicate more clearly, we have followed your advice and now have reorganized Figure 3 as follows:

[Figure]

Figure 3: The framework of the mass-conserving approach (MCMFE-NO$_x$).

Additionally, we have designated Response Figure 1 as Figure S1 and Response Table 1 as Table S1, and made the following changes in section 3.1:

"The sensitivity of the fitted coefficients ($\alpha_1$, $\alpha_2$, and $\alpha_3$) remain relatively stable to the changes in the a priori NO$_x$ emissions. In specific, we design a perturbation run in which the emissions are randomly altered day-by-day and grid-by-grid from the priori dataset near the extreme upper and lower bounds of their ±30% uncertainty range. This is then used in combination with the original values from TROPOMI to refit the coefficients, as given in Table S1 and Figure S1. It is observed that over 60% grids of the NO$_x$/NO$_2$ ratios and lifetimes, 40% grids in terms of the transports term are found to be robust, i.e., have a

change smaller than the 30% perturbed a priori emissions."

We hope this clarifies the issue and demonstrates the reliability of our approach. Please let us know if you have any further questions or require additional details.

2) It is unclear to me the locations of different sources used in Figures 4 and 5. An idea is to present maps of the derived monthly alpha1 (e.g., median values in representative months) overlaid by the different types of sources. Such maps will present an overview of combustion efficiency/types across the domain, and their association with the labeled emitters, greatly aiding the interpretation of Figures 4 and 5. Please consider adding such information in the revision or as an appendix.

Thank you for helping us to communicate the combustion efficiency ($\alpha_1$) more clearly.

As illustrated in Response Figure 2, the spatial distribution of various emission sources in Region 1 is presented. This region encompasses a higher concentration of industrial cities compared to the other two regions, with a more significant presence of various industrial facilities. Specifically, the grids in Response Figure 2 include 30 biomass burning facilities (green), 99 cement factories (yellow), 195 heat and production supply facilities (purple), 104 power plants (red), and 85 steel and iron factories (blue).

[Figure]

Response Figure 2. Map of the geographic locations of five different sources

To address your concern, we also present maps depicting the derived temporal median $\alpha_1$ values for grids overlaid with five

source types, as shown in Response Figure 3a-3e. Grids with $NO_x/NO_2$ ratios ranging from 1 to 5 are shaded green, those from 5 to 10 are shaded yellow, and those exceeding 10 are shaded red. These maps illustrate the spatial distributions of $NO_x/NO_2$ ratios from each source, and the statistical proportion of grids within these ranges is detailed in Response Table 2.

It is well-established that the $\alpha_1$ ratio is initially determined by the source type, thermodynamic conditions, and the availability of nitrogen, oxygen, and water vapor. Emissions rapidly adjust from the hot air emitted at the stack or pipe exit until reaching equilibrium in vertical height and thermodynamic conditions. This adjustment is observed at sites with similar climatological conditions, which affects the values of $\alpha_1$. However, the proportion of grids with $\alpha_1$ values exceeding 10 still exhibits a distinct differentiation among three groups: steel and iron factories (up to 52%), power plants (intermediate values, about 40%), and cement factories, heat production and supply, and biomass burning (lower values). It proves that the original thermodynamic conditions of combustion significantly influence the $\alpha_1$ values at the scales as observed by TROPOMI. These sites can therefore be differentiated by the emission source type and the observed $NO_2$ column loading (Figure 5 and Table 2).

Response Table 2. The account of grids from different sources over which the median values of $NO_x/NO_2$ is from 1 to 5, from 5 to 10 and above 10.

| Different sources (Number) | From 1 to 5 | From 5 to 10 | Above 10 |
|---|---|---|---|
| Biomass Burning (30) | 22% | 48% | 30% |
| Cement Factories (99) | 29% | 36% | 35% |
| Heat Production and Supply (195) | 34% | 42% | 24% |
| Steel and Iron Factories (85) | 24% | 24% | 52% |
| Power Plants (104) | 20% | 40% | 40% |

[Figure]

Response Figure 3. Map of median values in 12 months overlaid by the different types of sources: a) Biomass Burning, b) Cement Factories, c) Heat Production and Supply, d) Steel and Iron Factories; e) Power Plants.

Response Figure 4 also exhibits combustion efficiency and types from power plant sources, correlating these with the identified emitters. The four highlighted sites employ three main combustion technologies: Subcritical, Supercritical and Ultra-Supercritical, each exhibiting varying efficiency and heat rate. It is observed that, despite both sites employing Subcritical combustion technology (as depicted in Response Figure 4, Sites 2 and 3), their $NO_x/NO_2$ ratios differ due to different $NO_x$ control techniques. While the $NO_x/NO_2$ of Site 3 is 6.5, the Site 2 implements a more stringent control system, utilizing both Low $NO_x$ Burners (LNB) and Selective Catalytic Reduction (SCR), which likely contributes to its lower $NO_x/NO_2$ ratio (about 4). Supercritical (Site 4) and Ultra-Supercritical (Site 1) technologies demonstrate lower heat rates, which can be attributed to their higher combustion temperatures and efficiencies, potentially leading to increased $NO_x$ emissions. The $NO_x/NO_2$ ratio at Site 4 is approximately 12, whereas at the Site 1, it reaches a maximum of 26, corresponding with its highest efficiency and lowest heat rate. In conclusion, the $NO_x/NO_2$ ratio can partially reflect the characteristics of combustion efficiency and types. However, real-world conditions are considerably more complex, influenced by additional factors such as $NO_x$ control technologies (LNB and SCR), combustion technologies (related to the heat rates and efficiency), and local policies.

[Figure]

Response Figure 4. Map of median values in 12 months overlaid by Power Plants with three main combustion technologies: Subcritical, Supercritical and Ultra-Super.

It is hoped that the additions of Figures 4 and 5 will improve the interpretation by visually correlating the spatial distribution of alpha1 with the various emission sources. We have incorporated Response Figures 2-3 and Table 2 into the supplementary materials as Figure S2 and Table S2 to enhance the clarity and interpretability of our findings. Additionally, we have made the following changes in Section 3.1:

"The spatial distribution of five emission source types and their temporal median $NO_x/NO_2$ values in Region 1 are presented in Figure S2, with statistics of grids within different ranges of $NO_x/NO_2$ given in Table S2. The proportion of grids with $\alpha_1$ values exceeding 10 continues to exhibit a distinct difference between three groups: steel and iron factories (up to 52%), power plants (intermediate values, about 40%), and cement factories, heat production and supply, and biomass burning (lower values). Even though the emissions rapidly adjust from the hot air emitted at the stack or pipe exit, this is clearly significantly influenced by the thermodynamics of combustion itself, as well as additional factors including $NO_x$ control technologies (LNB and SCR), combustion technologies (related to the heat rates and efficiency) and local policies. These results demonstrate clearly that the original thermodynamic conditions still significantly influence the $NO_x/NO_2$ values at the scale observed by TROPOMI."

Thank you once again for your valuable feedback.

3) In Line 236, it is mentioned that emissions are monitored by the Continuous Emission Monitoring Systems (CEMS) network. However, no validation attempt is made to compare their derived $NO_x$ emissions against these ground truths. Please consider adding this important validation or clarify why it was not conducted.

Thank you for your valuable feedback. I apologize for the imprecise wording in Line 236, which led to a misunderstanding. My intention was to convey that all these locations (their geographical coordinates) of pollutant sources are tracked, as provided by the Pollutant Discharge Permit Management Information Platform of the Ministry of Ecology and Environment. However, it is important to note that not all these sources are large enough to be equipped with Continuous Emission Monitoring Systems (CEMS) to monitor their emissions. This is especially so for those plants which are small to medium in size or in terms of industrial production. However, all of them do possess discharge permits for pollutant gases including $NO_x$. Consequently, a comprehensive validation using CEMS data is not feasible for all the sources analyzed in our study. We would be very happy to make such a comparison with the fractions of those cites which do contain CEMS observations, however despite our efforts, we were unable to obtain the necessary permissions to access detailed CEMS data for the study period and locations of interest. If you have such data, we would be happy to directly compare.

To clarify, our study utilizes the location data to analyze the emission results and thermodynamic characteristics of various

sources, both of which are from our own model using satellite data from TROPOMI, rather than directly monitoring their emissions through CEMS. The location data allows us to correlate satellite observations with known emission sources, providing valuable insights into emission patterns and their thermodynamic properties.

We appreciate your understanding and will ensure this distinction is clearly communicated in the revised manuscript. We have updated section 2.6 accordingly:

"It is important to note that not all these sources are of sufficient scale to be equipped with Continuous Emission Monitoring Systems (CEMS) for emissions monitoring. Many of these sources are small to medium-sized industries, which do emit pollutant gases such as $NO_x$ and have applied for formal discharge permits. The location data enables us to correlate satellite observations with identified emission sources, thereby providing valuable insights into emission patterns and their thermodynamic characteristics."

4) Section 3.4 appears abruptly in the current manuscript. I suggest some motivation words regarding $NO_x$ emissions above water should be presented in the Introduction Section. In addition, although it is obvious, please quantitatively discuss how the estimated $NO_x$ emissions cannot be explained by river/lake emissions (e.g., comparing the magnitudes vs. typical emission numbers in the literature).

Thank you for your constructive feedback on Section 3.4.

We appreciate your suggestion to better motivate the context regarding $NO_x$ emissions above water bodies in the Introduction. And we acknowledge the importance of quantitatively discussing how the estimated $NO_x$ emissions cannot be solely attributed to river or lake biological or shipping sources of emissions. In the revised manuscript, we will include a comparison of our estimated $NO_x$ emissions with typical emission values reported in the literature for river and lake sources, as well as the thermodynamic ratio of $NO_x/NO_2$. There are few research papers specially studying on the $NO_x$ emissions over the Yangtze River, we still find one paper published in ACP (Zhang et al., 2023), working on the shipping emission inventory based on automatic identification system (AIS) combined with the China Classification Society (CCS).

As illustrated in the new figure 8 and table 3, the MCMFE-$NO_x$ values for all grids, as well as the proportion of MCMFE-$NO_x$ from power and industrial sources, are both higher than those in INTAC. This discrepancy indicates a potential underestimation of emissions from small and medium sources. In the SHIPPING sector of INTAC, $NO_x$ emissions account for less than 10% of the total emissions across all sectors, which is significantly lower than the estimates derived from the total ship emission based on AIS combined with CCS (Zhang et al., 2023). The study reports values as high as 83.5 Kt year$^{-1}$, which is 10 times greater than the shipping sector estimates from INTAC. It reveals that the values simulated in this work,

which include contribution from all the sectors are unlikely to be overestimated. Additionally, the $NO_x$ emissions over the Yangtze River Bridges in INTAC are considerably lower than those in MCMFE-$NO_x$, suggesting that heavy vehicular transport emissions in these specific grids are also underestimated.

We have reorganized the Figure 8 and Table 3 in the following way, including the comparison with INTAC-$NO_x$ (all sectors and the shipping sector) and the total ship emissions derived by (Zhang et al., 2023) in Jiangsu section:

[revised manuscript text omitted]

….."

Furthermore, we have made the following adjustments in the conclusion for clarity:

"This set of findings indicate that the contribution from small scale industries in pixels on or adjacent to rivers offer a significant source of unaccounted for $NO_x$ emission, which is shown to be larger than the amounts reported from biological sources on lakes (Kong et al., 2023) and inland shipping activities (Zhang et al., 2023)"

We believe these revisions will provide a clearer context and strengthen the overall narrative of our study. Thank you for your valuable suggestions.

Specific comments:

1) Please indicate a typical unit conversion factor from the estimated emissions at $\mu g/m^2/s$ to some more widely used unit (e.g., T/year) so that they are broadly understood and comparable.

Thank you for your valuable comment. The emission intensity, measured in $\mu g/m^2/s$, serves as an effective metric for comparing emissions across different cities, as it normalizes emissions by area, thereby facilitating direct comparisons irrespective of city size. For example, although Macao's total emissions (T/year) are projected to be very small due to its limited area, its emission intensity exceeds that of many larger cities. This is consistent with the known idea that Macao has a very high penetration of fossil-fueled vehicles and shipping, as well as fewer non-emitting sources of power generation, compared with many larger cities. This rationale supports our use of $\mu g/m^2/s$ in Section 3.2, as it aligns the derived emission intensities with the human activities and industrial characteristics of these cities or elucidates their developmental trajectories. However, in Section 3.4, where we compare the derived river emissions with other inventories and literature, we have opted to utilize Kt/year, which is a standard used more by the policy making community.

In response to your request for a more commonly used unit, such as T/year, we have included Response Table 3 to supplementary materials as Table S3 that details the 25th percentile, mean, and 75th percentile values of daily and grid-based emissions (T/day) for 30 cities across three regions (as listed in Figure 1). This supplementary material is intended to enhance clarity and facilitate easier comparisons for readers.

Response Table 3. The 25th percentile, mean, and 75th percentile values of day-by-day and grid-by-grid emissions (T/day) for 30 cities

| Region 1 | Cities | Mean | 25th | 75th |
|---|---|---|---|---|
| 1 | Beijing | 3.35 | 1.65 | 4.27 |

| | | | | |
|---|---|---|---|---|
| 2 | Tianjin | 4.70 | 2.64 | 5.70 |
| 3 | Tangshan | 5.01 | 2.63 | 6.02 |
| 4 | Jinan | 3.75 | 2.36 | 4.38 |
| 5 | Zibo | 3.81 | 2.26 | 4.55 |
| 6 | Jining | 3.29 | 2.28 | 3.90 |
| 7 | Heze | 2.90 | 2.04 | 3.51 |
| 8 | Qingdao | 2.79 | 1.74 | 3.44 |
| 9 | Xuzhou | 2.86 | 2.04 | 3.45 |
| 10 | Shijiazhuang | 3.76 | 2.28 | 4.64 |
| 11 | Xingtai | 3.55 | 2.39 | 4.26 |
| 12 | Handan | 3.76 | 2.39 | 4.38 |
| 13 | Lianyungang | 2.86 | 2.04 | 3.44 |
| Region 2 | Cities | Mean | 25th | 75th |
| 14 | Shanghai | 4.62 | 2.13 | 5.81 |
| 15 | Suzhou | 3.28 | 1.82 | 4.07 |
| 16 | Wuxi | 2.67 | 1.66 | 3.25 |
| 17 | Changzhou | 3.08 | 1.97 | 3.55 |
| 18 | Nanjing | 3.08 | 1.65 | 3.27 |
| 19 | Ma'anshan | 2.56 | 1.66 | 2.88 |
| 20 | Wuhan | 2.66 | 1.31 | 2.62 |
| 21 | Hangzhou | 1.74 | 0.89 | 2.07 |
| 22 | Ningbo | 2.03 | 1.16 | 2.27 |
| Region 3 | Cities | Mean | 25th | 75th |
| 23 | Xiamen | 2.73 | 1.50 | 3.40 |
| 24 | Shantou | 2.14 | 1.57 | 2.34 |
| 25 | Dongguan | 4.15 | 2.61 | 4.66 |
| 26 | Foshan | 4.11 | 1.98 | 4.74 |
| 27 | Hongkong | 4.38 | 2.17 | 5.07 |
| 28 | Macao | 2.98 | 1.78 | 4.14 |
| 29 | Guangzhou | 3.30 | 1.65 | 3.75 |
| 30 | Beihai | 1.46 | 1.07 | 1.58 |

2) In the revision, please use hyphen (-) in the words separated by two lines.

Thank you, it has been modified.

3) Line 45: change "This approach also includes..." to "This approach can also be applied to biomass burning emissions by including…".

Thank you again.

4) Line 55-58: Indeed, these existing approaches focus on monthly emissions. But evidences of information content (e.g., degrees of freedom of the framework) and validation (e.g, vs. daily CEMS emissions?) should be presented to justify the applicability of the proposed approach to estimate daily $NO_x$ emissions.

Thank you for helping us to communicate the results better. While existing approaches predominantly focus on monthly emissions, we recognize the necessity of validating our proposed method for daily $NO_x$ emissions. A more comprehensive sensitivity analysis has been conducted to assess the robustness of our methods, in a separate manuscript that we have recently submitted, which is currently under review (Lu et al., 2024).

In this manuscript under review at the present time, we have assumed an additional case in which the TROPOMI $NO_2$ observed column values include uncertainties at the extreme upper and lower bounds of their ±40% uncertainty range respectively, and use these to re-compute the fitting factors and subsequent emissions. When the TROPOMI $NO_2$ column values were adjusted, all related factors were simultaneously modified. This set of uncertainty simulations was uniformly applied as the $TO_{40\%}$ case, where the $NO_2$ columns were multiplied by random perturbations ranging from 0.6 to 1.4. The coefficients were refit using these new values from TROPOMI and the same values from both INTAC and meteorology over the entire domain included in this work. The results are provided in Response Figure 5 and Response Table 4.

The first point to note is that the refit values of the chemical and thermodynamic terms yield uncertainty ranges that are smaller than their respective perturbations, indicating that the fits are stable to the uncertainty perturbation. At the 40% uncertainty level, there is a reduction in short-term transport and an increase in long-term transport. This again is consistent with the uncertainty in the a priori emissions estimate. However, accounting for the net uncertainties (i.e., including the buffering effects of the chemical and thermodynamic terms with the more uncertain transport term) our overall findings demonstrate that the mass-conserving flexible emissions inversion method provides robust inversion results (as illustrated in

Response Figure 6). This is especially so when compared to the traditional wind speed and concentration gradient method, as observed in the transport term being the least stable to the uncertainty perturbation.

[Figure]

Response Figure 5. The distributions of three key coefficients obtained from $TO_{0\%}$ and $TO_{40\%}$: a) $NO_x/NO_2$, b) Lifetime [hours], c) Transport [km]

Response Table 4. The $20^{th}$, $50^{th}$, $80^{th}$ percentile ranges of three coefficients from $TO_{0\%}$ and $TO_{40\%}$.

| $NO_x/NO_2$ | $TO_{0\%}$ | $TO_{40\%}$ |
|---|---|---|
| 20% | 2.76 | 2.63 |
| 50% | 6.30 | 6.08 |
| 80% | 14.6 | 13.4 |
| Lifetime (days) | $TO_{0\%}$ | $TO_{40\%}$ |

| | | |
|---|---|---|
| 20% | 0.29 | 0.30 |
| 50% | 0.48 | 0.48 |
| 80% | 0.68 | 0.68 |
| Transport (km) | $TO_{0\%}$ | $TO_{40\%}$ |
| 20% | -323. | -403. |
| 50% | 18.1 | 24.0 |
| 80% | 351. | 416. |

[Figure]

**Response Figure 6**. The TROPOMI $NO_2$ column data is perturbated by a random factor 40% herein called [$TO_{40\%}$] to represent its range of uncertainty. The results are displayed for annual mean of emissions of a) $TO_{0\%}$, b) $TO_{40\%}$; and annual mean of error of c) $TO_{0\%}$, d) $TO_{40\%}$; and the annual mean of e) ($TO_{0\%}$-$TO_{40\%}$)/$TO_{0\%}$

It was observed that 93% of the daily grid cells exhibited a ratio of [($TO_{0\%}$-$TO_{40\%}$)/$TO_{40\%}$] within ±40%. The spatial and temporal 20th and 80th percentile ranges of $NO_x$ emissions for the $TO_{0\%}$ and $TO_{40\%}$ scenarios were 0.47 to 1.25 µg/m²/s and 0.45 to 1.25 µg/m²/s, respectively. Overall, the day-by-day and grid-by-grid $NO_x$ emission ranges are quite similar in both cases, as illustrated in Response Figure 7. On a month-by-month basis, the deviations for nearly all months are within ±10%

(February is excluded due to missing TROPOMI data in 2019 for region 2). January, October, and December exhibit the largest deviations, with $TO_{40\%}$ values consistently lower than $TO_{0\%}$ across all percentile ranges in January and October. Furthermore, the day-by-day spatial median values across different cases exhibit only slight variations. These findings indicate that changes in the driving factors ($\alpha_1$, $\alpha_2$ and $\alpha_3$) between the different $NO_2$ column loading scenarios are generally smooth and consistent, providing redundancy and being significantly influenced by the a priori emissions used in the fitting process. The constraints on the physically realistic values of $\alpha_1$ and $\alpha_2$, along with the constant use of INTAC, create a negative feedback loop affecting the relationship between $NO_2$ column changes and the final emissions products. This is consistent with the observed computed emissions and their differences.

[Figure]

**Response Figure 7**. a) The PDF of $NO_x$ emissions [μg/m²/s] over all individual days and grids of $TO_{0\%}$ and $TO_{40\%}$; b) The $20^{th}$, mean and $80^{th}$ values across different months of $TO_{0\%}$ and $TO_{40\%}$; c) The time series of the spatial median values of $TO_{0\%}$ and $TO_{40\%}$ for whole year.

The degrees of freedom of the framework are detailed in the supplementary materials, which provide a robust justification for the daily estimation approach. We have further identified Response Figure 6 and 7 as Figure S3 and Figure S4, with subsequent revisions in section 3.2:

"A comprehensive sensitivity analysis has been conducted to assess the robustness of MCMFE-$NO_x$ (Lu et al., 2024**)**. The degrees of freedom of the framework are detailed in the supplementary materials, which provide a robust justification for the daily estimation approach. A set of uncertainty simulations is uniformly applied as the $TO_{40\%}$ case, where the $NO_2$ columns are multiplied by random perturbations ranging from 0.6 to 1.4. By accounting for the buffering effects of the chemical and thermodynamic terms, our findings demonstrate that the mass-conserving flexible emissions inversion method yields robust inversion results (as presented in Figure S3) when compared to the traditional wind speed and concentration gradient method. It is observed that 93% of the daily grid cells exhibited a ratio [($TO_{0\%}$-$TO_{40\%}$)/$TO_{40\%}$] within ±40%. The day-by-day and grid-by-grid $NO_x$ emission ranges are quite similar in both cases (as presented in Figure S4). These findings indicate that changes in the driving factors ($α_1$, $α_2$ and $α_3$) across different $NO_2$ column loading scenarios are generally smooth and consistent."

5) Line 62 criticized the "inability to scale zero emissions" as a drawback of previous methods. However, Line 156-157 directly omit any locations with small MEIC emissions in the fitting. Is there a better way to resolve emission locations that are misrepresented by the a priori in your method?

The a priori dataset utilized (INTAC) contains a substantial number of extremely low emission values, specifically those below 0.3 μg/m²/s. We have reservations about the reliability of the fitted values in these regions, as they are too low to fall within the error range of the TROPOMI sensor, and previous models have struggled to address these areas effectively. Our approach integrates the fitting factors using a bootstrap method and TRPOMI observations, to compute the emissions. The distributions of the parameters $α_1$, $α_2$, and $α_3$ are utilized in Eq. 3 to compute the final emissions. This methodology facilitates a broader range of potential emissions datasets within the modeling framework, which may not be attainable through more heavily constrained or fitted modeling techniques. The day-to-day and grid-by-grid emissions are found to be 1.22±0.63 μg/m²/s on pixels in which the a priori inventory is lower than 0.3 μg/m²/s.

6) Line 78-79: "The results are checked against independent measurements of NO$_x$ emissions flux". Where in the paper?

Thank you for pointing this out. We have corrected our explanation of the CEMS data above. Since we were not able to access their observations directly, we now propose removing this from the paper. If you happen to have access to this data, please let us know, and we will include it in the updated version.

7) Figure 1: While Regions 2 and 3 seem representative to me, please explain why some large and industrialized cities in the nearby Henan and Shanxi provinces are excluded in Region 1?

The selection of Region 1, which includes Beijing, Tianjin, Hebei Province, substantial portion of adjacent Shandong Province, Xuzhou, and Lianyungang in Jiangsu Province, was based on several key considerations: 1) The chosen areas form a contiguous geographical region with significant administrative and economic interconnections. The boundary we selected aligns with a contiguous contour of high NO$_2$ climatological loading, facilitating a more integrated analysis of atmospheric conditions and emissions patterns, thereby reducing the influence of observational noise. The areas in Jiangsu between Xuzhou and Lianyungang and those in Region 2 are separated by relatively low polluting areas and therefore form a clear boundary in terms of geospatial distribution of emissions. 2) While Henan and Shanxi provinces also contain large industrialized cities, the regions included in Region 1 already encompass a diverse range of urban and industrial environments. This diversity effectively represents the broader characteristics we aim to study, negating the need to extend the boundaries further. 3) Most importantly, the elevation of Region 1 is predominantly below 500 meters, with nearly 94% of the spatial regions falling within this range. In contrast, Shanxi Province has less than 4% of its area below 500 meters, and Henan Province also exhibits relatively higher elevations. Furthermore, there are high mountains separating these regions in Shanxi and Henan from the other regions analyzed herein. These elevation differences significantly influence wind patterns, as higher elevations can lead to distinct wind dynamics (Pepin et al., n.d.). The issues concerning canyon effects and upslope/downslope winds are complex and best left for ultra-high resolution future studies. For this study, we selected wind data at 950 hPa to ensure consistency and relevance to the elevation profile of Region 1, and for this reason have excluded the areas as given.

8) Line 145: please clarify what weights (surface area?) are used in the re-gridding.

Thank you for your question. In our re-gridding process, we utilized the weighted polygon shaped remotely sensed measurement toolkit HARP (Harmonized Atmospheric Retrievals for Pollution) which uses an area-weighted methodology. Specifically, the pixels of NO$_2$ column observations within each swath are amalgamated into unified latitude-longitude grids measuring 0.05° x 0.05° in size. The weights are calculated based on the surface area of the grid cells, ensuring that the regridded values accurately represent the spatial distribution of the original data. This method allows for precise interpolation and maintains the integrity of the spatial patterns in the atmospheric data. We have revised the relevant parts of 2.2 in the following way:

"As shown in Figures 2a and 2b, the pixels of $NO_2$ column observations within each swath are amalgamated into unified latitude-longitude grids measuring 0.05° x 0.05° in size, using the weighted polygon shaped remotely sensed measurement toolkit HARP (http://stcorp.github.io/harp/doc/html/index.html). An area weighted average is performed, ensuring that the re-gridded values accurately represent the spatial distribution of the original data (http://stcorp.github.io/harp/doc/html/algorithms/regridding.html)."

9) Line 149: Higher resolution (e.g., 1-km) emissions inventory is available in China (e.g., https://doi.org/10.1016/j.scib.2020.12.008).

Thank you for pointing this out. We have now acknowledged this higher resolution emissions inventory at 1-km resolution, and include the references behind it. However, we need to be very clear. This current paper, to our understanding, offers 1 data point per grid per year, which would make it a single annual average value, and it does not provide a robust uncertainty range on that datapoint. Furthermore, it is based on data from 2013. In contrast, our study employs the INTAC inventory from 2017, which provides high resolution temporal data (12 data points per grid), which is closer to the observed data from TROPOMI $NO_2$ VCDs in 2019. Changes to the text include:

"The original INTAC emissions are quantified in units of Mg/grid/month, with a temporal resolution of one month and a spatial resolution of 0.1° x 0.1° for 2017. It is important to note that a higher spatial resolution inventory, the 1-km resolution by MEIC is also available (Zheng et al., 2021). However, this 1-km inventory only offers one data point per grid per year, while also providing insight into emissions from 2013, which in China are quite different from those in 2019. For this reason, we have used the INTAC inventory herein since it more closely matches the 2019 TROPOMI data. Both datasets include coverage over mainland China and include emissions from eight sectors: power, industry, residential, transportation, agriculture, solvent use, shipping, and open biomass burning (Wu et al., 2024)."

10) Line 160-165: the ERA5 wind is at 0.25 degree resolution. How would the averaging-out of the wind fields affect the divergence calculation at 0.05 degree?

The specific ERA5 product used in our study is obtained at a spatial resolution of 0.25° x 0.25°. This data is subsequently interpolated onto the TROPOMI grid of 0.05° x 0.05°. However, our research regions are significantly larger than the 0.25° x 0.25° scale. Consequently, the relative patterns and trends remain comparable, even if some fine-scale details may be

averaged out. Most importantly, the ERA5 data is widely recognized for its comprehensive temporal and spatial coverage, as well as its high-quality reanalysis products. Most studies and researches in this field employ ERA5 data, underscoring its reliability and acceptance within the scientific community. Given the lack of higher-resolution alternatives that offer similar coverage and quality, ERA5 remains the most suitable choice for this study. If there were very high-resolution meteorological datasets available day-by-day throughout the entire year in the public domain, we would gladly use them and perform various additional sensitivity tests. Please let us know.

11) Line 215: please explain why only alpha-1 is bootstrapped? Is it considered as the dominant contributor to the uncertainty of the system?

We appreciate the reviewer's question regarding the bootstrapping of only the $\alpha_1$ parameter. The decision to bootstrap $\alpha_1$ was based on its significant role in the system's uncertainty and its thermodynamic relevance. However, the resulting sampling of $\alpha_2$, and $\alpha_3$ are also sampled. It is important to maintain the physical connections between the terms. Bootstrapping $\alpha_1$, $\alpha_2$, and $\alpha_3$ separately would disrupt the physical interconnections among these factors. If one were to randomly sample them independently of each other, then the physical meaning of the conservation and the underlying factors would be lost. Then the situation would merely become a mathematical problem.

We have also tested bootstrapping only $\alpha_2$ and only $\alpha_3$ and then using the respective other dependent values. The end results are similar. Considering computational constraints and the necessity to prioritize key parameters, it was determined that bootstrapping $\alpha_1$ along with its associated terms would be the most effective strategy.

Furthermore, this follows the procedures for running model ensembles that have been established by the modeling community. When running such models, only certain independent variables can be randomly sampled in groups, while others are not which are dependent on the randomly sampled variables.

12) Line 251-252: this observation is consistent with the very dense distribution of strong $NO_x$ emission sources in China (whereas sources are more distant in the US and Europe), as will be presented in Figures 6 and 7. It is suggested to synthesize these findings in the discussion.

We appreciate the reviewer's suggestion to synthesize the findings related to the spatially dense distribution of strong $NO_x$ emission sources in China in the discussion. The observation that a significant amount of transport into emitting areas comes from upwind emitting areas is indeed consistent with the computed positive transport (net import) values of 45%, 51%, and 46%. This pattern aligns with the dense distribution of $NO_x$ sources in China, as opposed to the more dispersed sources in the US and Europe. This is clearly observed in the maps above, in which there are multiple sources found in adjacent grids

(at a resolution of 0.05º x 0.05º) sometimes stretching continuously over 1º or more.

To address this, we propose to include a synthesis of these findings in the discussion section, highlighting the following points:

The high geospatial density of $NO_x$ emission sources in China, when coupled with the lifetime of $NO_x$ in-situ, leads to significant transport of emissions from upwind sources into other downwind emitting areas. This results in higher net import values, as observed in our study. This means that the commonly used assumption including clean background, constant background, and background subtraction will yield an unreasonable answer in the context analyzed herein. This is especially true given that such continuous sources are shown to exist over 10's to 20's of boxes in a row in some regions herein. In contrast, $NO_x$ sources in the US and Europe tend to be more widely spaced, leading to different transport dynamics, and lower net values imported into emitting regions from upwind. This distinction underscores the importance of a precise regional emission source distribution in both space and time being required to understand transport patterns.

We will reference Figures 6 and 7 to visually support these findings, providing a clear comparison of $NO_x$ source distributions and their impact on transport dynamics. By synthesizing these observations in the discussion, we aim to provide a comprehensive understanding of the transport dynamics and their relation to emission source distributions in different regions.

We have rephrased this part in the following manner:

"Overall, the community has assumed that negative transport, or net export from highly emitting boxes, dominates the transport. In specific, it has generally been assumed that emissions exit from an urban area, and the impact of upwind sources entering into the background of an urban area or source are frequently not considered. However, the results herein show that this is actually only the case 55%, 49%, and 54% of the time over the three domains respectively. This means that a significant amount of mass is transported into emitting areas from upwind emitting areas, and is consistent with the computed positive transport (net import) values of 45%, 51%, and 46%. There are some theoretical studies and case studies which have demonstrated that this is the case, but none have used observations over such a long time period to analyze the frequency of occurrence (Cohen et al., 2011; Cohen and Prinn, 2011; Wang et al., 2023)."

13) Line 263: It seems to me that distributed residential heating might have relatively lower $NO_x/NO_2$ because of the relatively lower temperature and exposure to ambient ozone, than large point sources like power plants. Is the heating in North China somewhat different?

You are correct that distributed residential heating systems generally produce lower $NO_x/NO_2$ emissions compared to large point sources, such as power plants. This difference is primarily attributed to the lower combustion temperatures and reduced exposure to ambient ozone in residential heating systems. However, as indicated in Line 263 and presented in Figure 4c, this paragraph focuses on the comparison of $NO_x/NO_2$ across the three regions, which encompass large-scale sources throughout entire regions. Therefore, these regions are not directly comparable to individual power and industrial sources. In Figure 5, we compare the emissions from specific sources with one another, and it is evident that the 80[th] percentile of $NO_x/NO_2$ emissions from power plants is indeed higher than that of the whole Region 1.

Regarding the heating systems in North China, there are indeed significant differences. In urban areas, centralized heating systems, including combined heat and power (CHP) plants and heating-only boilers, are prevalent (Liu et al., 2024). These systems typically operate at higher combustion temperatures and for extended periods, which can lead to higher $NO_x$ emissions when compared to distributed residential heating. In rural areas, decentralized small coal stoves are commonly used, which also contribute to $NO_x$ emissions but at different rates and patterns, in both space and time, when compared to centralized systems. The observed secondary peak in February for Region 1 can be attributed to the increased demand for heating during the colder months, which is consistent with the operational patterns of centralized heating systems in North China (Yuan et al., 2024). These systems usually remain active until early March, leading to higher emissions during this period.

In addition, these differences in space and time also reflect the large impact on economic and energy demands during the Chinese New Year period in Northern China.

I hope this addresses your concerns. Please let me know if further clarification is needed.

14) Line 270-273: Overall there is very weak seasonal variation in the derived lifetimes. This is in contrast to other studies where the $NO_x$ lifetime in summer and winter can differ by a factor of 4 (e.g., https://doi.org/10.5194/acp-20-1483-2020). How should we interpret these differences?

Thank you for your comment. The differences in seasonal variation of $NO_x$ lifetimes between our study and others, such as the one you referenced (Shah et al., 2020), can be attributed to several factors as follows. We of course will cite this and other important and relevant literature. Thank you for point this out!

First, our study focuses on day-to-day results, and therefore provides a greater amount of variability, as compared to seasonal averaging. There is also a sampling issue here: when producing seasonal averages, how are missing data and/or days considered? Are extremes in the temperature profiles and changes in emissions considered at high resolution? To work to

overcome these issues, I would invite the modelers to compute day-to-day results and uncertainty bounds, and then compare these with our day-to-day results herein. We would be happy to receive such feedback for us to improve.

Second, the geographical and temporal scope can significantly influence the observed $NO_x$ lifetimes due to variations in local meteorological conditions, emission sources, and atmospheric processing. This study looks grid-by-grid at high resolution spatially, and therefore does not lead to as much smearing between high and low emissions grids co-located within the same larger grid. The lifetime is a function of the chemical decay, as well as diffusion between that grid and adjacent grids, changes in the UV profile, local changes in OH and temperature, dynamics, and many other factors impacting a first order decay. This is a complex analysis which should be looked at using high resolution processes.

Next, we observe that that paper cited employs a chemical model, GEOS-CHEM. We know that the actual lifetimes of reactions have considerable uncertainty and variability when they are derived (Seinfeld et al., 1998). Furthermore, we know that the reactions are highly sensitive to clouds, aerosols, and other in-situ species? On top of this, we have found observational data which shows that high values of a1 actually do exist in urban areas as observed in this work (Karl et al., 2023). A very interesting question would be to compare the results and uncertainties herein with the model results and uncertainty ranges of the lifetimes at high frequency. We are happy to work together to address this important task as future work.

What we have demonstrated is that the median values of $NO_x$ first order decay lifetime range from 10.4 to 12.4 hours, 9.9 to 12.6 hours, and 6.2 to 11.7 hours in different months for Regions 1, 2, and 3, respectively. The 10th and 90th percentile values are 3.8 and 18.8 hours, 4.6 and 20.6 hours, and 3.8 and 18.8 hours for Regions 1, 2, and 3, respectively. These results are based on total column values, which include temperature, UV, climate, and aerosols observed in the research regions.

Based on our findings (presented in Figure 4a), the largest $NO_x$ lifetime values are observed in November or December, while the smallest values are found in June (similar to the paper provided). Besides, over grids of 20th percentile values, the lifetime ranges from 5.4 hours (May) to 9.1 hours (November), align well with the changed in the paper cited. However, the differences between the lifetime of the 80th percentile become less pronounced, ranging from 14 hours (September) to 16.1 hours (April). This is due to the complex local conditions and forcing factors.

Our work demonstrates that November behaves more similarly to December than December does to February. We note with interest that the UV radiation in November is more similar to that in December than it is to February.

I hope this clarifies the differences. Please let me know if further explanation is needed.

15) Figure 4d: Please adjust the labels in the x-axis to align correctly with each bar. For example, the distances between Wuxi and Changzhou (or between Beihai and Hongkong) are somewhat too large?

Thank you for your comment. The Figure 4 has been replotted with same distance between each city:

[Figure]

Figure 4: The distribution (mean values, 20[th], 40[th], 60[th], and 80[th] percentile values) of monthly (a) lifetime; (b) NO$_x$/NO$_2$; (c) 20[th] and 80[th] percentile values of monthly NO$_x$/NO$_2$ in three regions; (d) Mean values of NO$_x$/NO$_2$ over 30 cities

16) Section 3.1: Besides the 5 sources discussed here, what is the typical range of NO$_x$/NO$_2$ for vehicle emissions? Can this information help explain the low values of cities like Macao (should be dominated by vehicle emissions)?

Thank you for your insightful thoughts. Vehicle emissions, particularly from gasoline and diesel engines, are significant sources of NO$_x$, which includes both NO and NO$_2$. However, computing the range of NO$_x$/NO$_2$ for vehicle emissions is challenging, as they are not as substantial as emissions and tend to be more intermixed on a grid-by-grid basis with other power and industrial sources.

To attempt to address this problem, we take a statistic of the NO$_x$/NO$_2$ over the Yangtze River Bridges in Nanjing, Wuhu, Anqing, Jiujiang, Huanggang and Wuhan, most of which exceed 5 km in length. The 20[th] and 80[th] percentiles of NO$_x$/NO$_2$ for transport activities over these bridges ranges from 5 to 15, which is consistent with findings from the existing literature indicating that the NO$_2$/NO$_x$ for vehicle emissions typically falls between 15% to 18% (Karl et al., 2023). In contrast, the

NO$_2$/NO$_x$ from power plants and industrial sources can be as low as 2% (Ma et al., 2016), as presented in Figure 5. In Macao, the NO$_x$/NO$_2$ ranges from 3 to 12, which seems to indicate that in fact transport emissions constitute a significant amount of the source. This seems consistent with the assertions made about transport being one of the most significant sources. This is also consistent with studies have confirmed that emissions from motor vehicles are significant in Macao (Xu et al., 2023), and that advances in hybrid electric vehicles (HEVs) and the implementation of specific NO$_x$ reduction devices in new cars have led to a substantial reduction in NO$_x$ emissions in Macao for recent years (Wu et al., 2015).

17) Line 311: Why does this NO$_x$/NO$_2$ ratio of power plant sources increase with more power plant emissions (activities)? Is it because your derived values are still contaminated by other emission sources for these grids (thus these July months are more dominated by power plants)?

Line 311: "However, July of the heat production and supply is an exception with high values as the hottest time of the year when extreme numbers of people turn on the AC."

We appreciate the reviewer's question regarding the NO$_x$/NO$_2$ ratio and its relation to power plant emissions. However, it appears there may be a misunderstanding. The content in Line 311 refers to "heat production and supply", not specifically to "power plants." Our point is that the high values observed in July are due to the increased demand for air conditioning during the hottest time of the year. Due to the increase in the temperature of cooling water during this time of the year, there is a significant impact on the efficiency of heat production and supply plants.

The heat production and supply encompass a broader range of activities, including district heating and cooling systems, which are heavily utilized during extreme weather conditions. This is distinct from power plants, which primarily generate electricity. In July, the extreme number of people using air conditioning leads to a substantial increase in heat production and supply activities. This heightened demand results in higher NO$_x$ emissions from these sources, which is reflected in the observed NO$_x$/NO$_2$ ratios.

We hope this clarification addresses the reviewer's concern and provides a clearer understanding of the factors influencing the NO$_x$/NO$_2$ ratios in our study.

18) Table 2: Very interesting classification method. Can this be used to categorize sources of grid cells where the dominant source is unknown (e.g., on the maps of alpha1)?

We acknowledge the potential value of this approach and propose it as an area for future research. We believe that it may be used for such purposes, but at the present time are working on adding in information from other co-emitted species (Cohen et

al., 2024). Future studies could explore the feasibility and effectiveness of using this classification method to categorize emission sources in grid cells with unknown dominant sources, thereby enhancing our understanding of emission patterns. This could be very interesting when applied to regions in which there is little a priori information about different source types. Thank you for point this out.

19) Line 348-357: It is suggested to make a table summarizing these numbers, and only highlights several of them in the main text.

Thank you for your valuable comment. In the main text, we have highlighted three or four typical cities in each region, including 13 cities in total. This will ensure that the main points are clearly communicated without overwhelming the reader with too many details. And we have made a table with a more widely used unit, detailing the 25th percentile, mean, and 75th percentile values of day-by-day and grid-by-grid emissions (T/day) for 30 cities in three regions (as listed in Figure 1). It comprehensively summarizes the numerical data and provides a clear and organized overview of the values, making it easier for readers to interpret the results, added in the supplementary material.

Response Table 3. The 25$^{th}$ percentile, mean, and 75$^{th}$ percentile values of day-by-day and grid-by-grid emissions (T/day) for 30 cities

| Region 1 | Cities | Mean | 25th | 75th |
|---|---|---|---|---|
| 1 | Beijing | 3.35 | 1.65 | 4.27 |
| 2 | Tianjin | 4.70 | 2.64 | 5.70 |
| 3 | Tangshan | 5.01 | 2.63 | 6.02 |
| 4 | Jinan | 3.75 | 2.36 | 4.38 |
| 5 | Zibo | 3.81 | 2.26 | 4.55 |
| 6 | Jining | 3.29 | 2.28 | 3.90 |
| 7 | Heze | 2.90 | 2.04 | 3.51 |
| 8 | Qingdao | 2.79 | 1.74 | 3.44 |
| 9 | Xuzhou | 2.86 | 2.04 | 3.45 |
| 10 | Shijiazhuang | 3.76 | 2.28 | 4.64 |
| 11 | Xingtai | 3.55 | 2.39 | 4.26 |
| 12 | Handan | 3.76 | 2.39 | 4.38 |
| 13 | Lianyungang | 2.86 | 2.04 | 3.44 |
| Region 2 | Cities | Mean | 25th | 75th |

| | | | | |
|---|---|---|---|---|
| 14 | Shanghai | 4.62 | 2.13 | 5.81 |
| 15 | Suzhou | 3.28 | 1.82 | 4.07 |
| 16 | Wuxi | 2.67 | 1.66 | 3.25 |
| 17 | Changzhou | 3.08 | 1.97 | 3.55 |
| 18 | Nanjing | 3.08 | 1.65 | 3.27 |
| 19 | Ma'anshan | 2.56 | 1.66 | 2.88 |
| 20 | Wuhan | 2.66 | 1.31 | 2.62 |
| 21 | Hangzhou | 1.74 | 0.89 | 2.07 |
| 22 | Ningbo | 2.03 | 1.16 | 2.27 |
| Region 3 | Cities | Mean | 25th | 75th |
| 23 | Xiamen | 2.73 | 1.50 | 3.40 |
| 24 | Shantou | 2.14 | 1.57 | 2.34 |
| 25 | Dongguan | 4.15 | 2.61 | 4.66 |
| 26 | Foshan | 4.11 | 1.98 | 4.74 |
| 27 | Hongkong | 4.38 | 2.17 | 5.07 |
| 28 | Macao | 2.98 | 1.78 | 4.14 |
| 29 | Guangzhou | 3.30 | 1.65 | 3.75 |
| 30 | Beihai | 1.46 | 1.07 | 1.58 |

We have incorporated Response Table 3 into the supplementary materials to enhance the clarity and interpretability of our findings, with the paragraph of section 3.2 reorganized in the following way:

"The daily average emissions and uncertainties of these selected representative urban areas are computed as follows: In region 1, Beijing, Tianjin, and Tangshan which are primarily coal-based and oil-based resource areas have values of $1.6\pm0.8$ $\mu g/m^2/s$, $2.3\pm1.0$ $\mu g/m^2/s$, and $2.4\pm1.1$ $\mu g/m^2/s$, respectively. Jinan and Zibo, rapidly industrializing locations, have emissions of $1.7\pm0.9$ $\mu g/m^2/s$ and $1.8\pm0.8$ $\mu g/m^2/s$. In region 2, $NO_x$ emissions in Shanghai are high at $2.0\pm0.5$ $\mu g/m^2/s$. Cities like Nanjing, Suzhou, and Wuhan, which have experienced rapid economic development, show values of $1.4\pm0.6$ $\mu g/m^2/s$, $1.5\pm0.7$ $\mu g/m^2/s$, and $1.2\pm0.5$ $\mu g/m^2/s$, respectively. Ma'anshan, with a rapidly developing industry, also has high emissions at $1.2\pm0.5$ $\mu g/m^2/s$. In region 3, cities near the Pearl River estuary engaged in wharf ship movement, such as Hong Kong, has emissions of $1.8\pm0.7$ $\mu g/m^2/s$. Cities like Dongguan and Foshan, which have undergone significant industrialization, show emissions of $1.7\pm0.6$ $\mu g/m^2/s$ and $1.7\pm0.5$ $\mu g/m^2/s$. The 25th percentile, mean, and 75th percentile values of daily and grid-based emissions (T/day) for 30 cities across three regions (as listed in Figure 1) are detailed in Table

S3."

Thank you again.

20) Line 366-373: Do you mean "urbanization" when referring to "land surface changes"? I do not foresee a clear connection between rapid "land changes" and "emission changes".

Thank you for your insightful question. In the context of our study, "land surface changes" encompasses a broader range of alterations than just urbanization. While urbanization is a significant component, we also consider other forms of land use and land cover changes, such as deforestation, agricultural expansion, and infrastructure development. These changes can significantly impact $NO_x$ emissions through various mechanisms. Urbanization, for instance, typically leads to increased vehicular traffic and industrial activities, both of which are major sources of $NO_x$ emissions. Additionally, the transformation of natural landscapes into urban areas can alter local meteorological conditions, such as temperature and wind patterns, which in turn affect the dispersion and concentration of pollutants. Similarly, agricultural expansion often involves the use of fertilizers, which can release $NO_x$ through soil microbial processes. Deforestation and infrastructure development can also contribute to changes in emission patterns by altering the natural carbon and nitrogen cycles.

Our intention was not to establish a direct causal link between rapid land changes and emission changes. Instead, we observed that the regions depicted in the Figure 6d-6f, where uncertainty is comparable to or exceeds day-to-day variability, are experiencing land-use changes. These regions are typically located in suburban or rapidly developing rural areas. This observation suggests that the current retrieval algorithms may not be sufficiently robust in these areas to yield trustworthy emissions inversions, potentially due to the complex and dynamic nature of land-use changes. Therefore, our intention is to propose that more robust validation and retrieval algorithms are necessary to accurately capture the emissions in these regions.

We have reorganized the third paragraph of section 3.2 in the following way:

"There is minimal overlap between regions with high day-to-day variability and regions with high uncertainty. In Wuhan, for example, high variability is observed in the city center, while high uncertainty is located north of the city near the river area. Regions shown in the Figure 6e where uncertainty is similar to (less than 0.5 µg/m²/s lower) or exceeding 0.3 µg/m²/s higher than day-to-day variability are undergoing land-use changes, indicating that more robust validation and retrieval algorithms may be required in these regions. The forms of land use and land cover changes, such as urbanization deforestation, agricultural expansion, and infrastructure development, can significantly impact $NO_x$ emissions through various mechanisms. These regions encompass the southern part of Hebei province, industrializing locations in Shandong province,

suburban areas around Xuzhou, Suzhou, Wuxi, Changzhou, Zhenjiang, and Nanjing in Jiangsu province, the northern expanded part of Wuhan, developing cities in Guangdong province and Xiamen. They are situated in suburban or rapidly developing rural areas that were previously overlooked by the a priori datasets, covering 22%, 24%, and 12% of region 1, 2 and 3, respectively. In contrast, Figure 6d illustrates that many metropolitan areas such as centers of Beijing, Tianjin, Shanghai, Hong Kong, Guangzhou, Suzhou, Changzhou, Nanjing, Hangzhou, Wuhan, and Xuzhou where land surfaces are not changing significantly, exhibit over 0.5 $\mu g/m^2/s$ smaller uncertainty than day-to-day variability. These grids cover approximately 6%, 5%, and 2% in region 1, 2 and 3, respectively. This study highlights the importance of considering day-to-day variability in emission calculations for these areas, emphasizing the limitations of relying on monthly or annual averages from a small sample of daily data. Additionally, this research includes a comparison of the monthly mean of uncertainty and monthly variability of emission, illustrated in Figure 6f."

21) Figure 7 and associated discussion: It looks to me that Figure 7b mostly highlights locations with strong $NO_x$ emissions (e.g., urban centers), and Figure 7a contains their downwind locations. Can part of this be explained by the transport (smearing) effects that are not fully accounted for by the inversion?

Thank you for your insightful observation regarding Figure 7. The main point is that MEIC does not necessarily have the correct spatial location for where the emissions occur. When working with a business, the address of the office and the address where the emissions occur may not and frequently do not match exactly. Attempting to extrapolate household and other small sources over distributed, moving, and rapidly economically increasing households and small enterprises is challenging. Furthermore, as explained above, due to the uncertainty inherent in the $NO_2$ column observations, we have excluded all pixels with emissions below a low cutoff value, so that we do not attempt to train emissions using noisy data. However, it is also important to note that the opposite is true: the pixels with the highest emissions in the MEIC inventory are also not found in our resulting emissions. We find similarly that certain ultra-high emissions pixels tend to not exist, or are shifted in space and time.

You mention the issue of smearing. While our inversion method includes some consideration for transport processes, the complexity and variability of atmospheric dynamics mean that some smearing effects are inevitable. However, as illustrated in Response Figure 5, the gradients over urban centers and downwind locations are relatively smooth. For instance, in Beijing, there are minor gradient variations between urban center pixels and surrounding areas, yet strong emissions persist in the northern part of the city.

[Figure]

Respond Figure 5: Annual daily mean of gradient of wind multiplied by TROPOMI $NO_2$ column loading $[\mu g/m^2/s]$

There are of course limitations. The gradient approach used herein is limited by missing pixels of data, which could lead to additional smearing on a day-by-day basis. There are some low-emitting regions which may actually be real, and may occasionally in time be retrievable by the method, but which we exclude. For these reasons, our emissions are conservative and like are an underestimate. However, our values are already larger than existing inventories.

We believe that investigation by models into day-to-day changes in their emissions may be a way to move this forward with respect to future research, and invite the community to use our products to drive their models. We are grateful for any feedback that these future studies may be able to provide, so that we can continue to improve.

22) Line 461: Please clarify why UV radiation is related with alpha1 ($NO_x/NO_2$).

As we have responded above, UVAI is a measurement that provides information on the column loading of absorbing aerosol in the UV. This absorbing aerosol in turn impacts the radiative flux at all visible and UV bands through absorption, scattering, and extinction. There has been extensive work which has demonstrated that the absorbing aerosols reduce the actinic flux and alter OH. In both cases, this has an impact on the atmospheric lifetime of $NO_x$. We agree that this is not the only component, but we do believe that it is fair to say that UVAI has an impact on the net actinic flux at the surface. This is also consistent with the paper you have mentioned above, in terms of the differences in the comparisons of the two $NO_2$ column observation products and how they diverge under different aerosol conditions, in specific that the POMINO product seems to work better under more complex aerosol conditions than the standard product.

Reviewer #2: This study introduces a novel mass-balance based analytical framework designed to rapidly and flexibly quantify $NO_x$ emission fluxes with high spatial and temporal resolution, using daily observations from TROPOMI across three rapidly changing regions in eastern China. The study effectively quantifies source attribution by revealing unprecedented insights into $NO_x/NO_2$ emission ratios across five industrial sources. The results also indicate significant discrepancies between observed emissions and those predicted by the MEIC a priori inventory. Furthermore, the paper emphasizes the substantial $NO_x$ emissions linked to small and medium industrial and residential activities in regions adjacent to rivers. This paper stands as a well-executed and valuable piece of research which is overall well-written. The methodology is emerging and innovative, and the results contribute valuable insights into the emissions landscape, there are certain critical details that require clarification. I look forward to re-reviewing the paper in the next round, provided that the following comments can be addressed.

Major comments:

1) The paper presents fitted alpha1-alpha3 values, but it is important to assess how sensitive these values are to variations in the initial assumptions about $NO_x$ emissions. Did the authors conduct a sensitivity analysis to determine if the alpha values remain stable under different scenarios? This information would provide insight into the reliability of the emissions estimates and the overall methodology.

We note that this is identical to main comment 1 from Reviewer #1. For the sake of clarity and thoroughness, we have repeated the detailed response here:

To address your concern about the robustness of our approach, we conducted sensitivity analyses to verify that the fitted coefficients (alpha1-alpha3) are relatively stable even with imperfect assumptions surrounding the a priori $NO_x$ emissions used in the first step.

The priori emissions dataset (INTAC) utilized in this study was developed by the MEIC team in collaboration with various authoritative research institutions. We consider it the most suitable choice for priori datasets over China. The uncertainties in INTAC arise from its integration process. Relatively low uncertainties are exhibited by $NO_x$ emissions of INTAC, benefiting from well-established estimates for large-scale combustion sources. Given this, we assume that the uncertainty is less than 30%. In this study, we selected Region 2 and tested what impacts would occur if the prior emissions were to have uncertainties near the extreme upper and lower bounds of their ±30% uncertainty range. The coefficients were refit using new priori emissions values ($INTAC_{30\%}$ Case) while maintaining the same values from TROPOMI. The results are presented

in Response Figure 1. It is observed that for most grids (over 60%), the $NO_x/NO_2$ ratios and lifetimes exhibit significant robustness following the 30% perturbation of INTAC as the priori. However, the transport parameter demonstrates less robustness, with approximately 40% of grids showing a lower ratio than the 30% perturbation, meaning slightly over half of the total grids may suffer from a non-linear response. As illustrated in Response Table 1, the 20th and 80th percentile ranges of $NO_x/NO_2$ are from 2.8 to 14.6 and from 2.7 to 15.0 in the $INTAC_{0\%}$ and $INTAC_{30\%}$ cases, respectively. The corresponding ranges for lifetime (days) are from 0.29 to 0.68 and from 0.27 to 0.73. The differences in transport (km) are slightly larger than those of the other coefficients, ranging from -323. to 351. and from -289. to 314. Our findings indicate that the variations in the fitted coefficients are relatively minor, thereby confirming the robustness of our method. And any slightly larger changes in the transport term are buffered by the non-transport terms.

[Figure]

Response Figure 1. The distributions of three key coefficients obtained from $INTAC_{0\%}$ and $INTAC_{30\%}$: a) $NO_x/NO_2$, b) Lifetime [hours], c) Transport [km]; The ratio of $(INTAC_{30\%}-INTAC_{0\%})/INTAC_{0\%}$ is also displayed on: d) $NO_x/NO_2$, e) Lifetime [hours], f) Transport [km];

Response Table 1. The $20^{th}$, $50^{th}$, $80^{th}$ percentile ranges of three coefficients from $INTAC_{0\%}$ and $INTAC_{30\%}$.

| $NO_x/NO_2$ | $INTAC_{0\%}$ | $INTAC_{30\%}$ |
|---|---|---|
| 20% | 2.76 | 2.68 |
| 50% | 6.30 | 6.24 |

| 80% | 14.6 | 14.9 |
| --- | --- | --- |
| Lifetime (days) | INTAC$_{0\%}$ | INTAC$_{30\%}$ |
| 20% | 0.29 | 0.27 |
| 50% | 0.48 | 0.48 |
| 80% | 0.68 | 0.73 |
| Transport (km) | INTAC$_{0\%}$ | INTAC$_{30\%}$ |
| 20% | -323. | -289. |
| 50% | 18.1 | 16.3 |
| 80% | 351. | 314. |

We have designated Response Figure 1 as Figure S1 and Response Table 1 as Table S1, and made the changes in section 3.1. The specific changes in the manuscript can be found in our response to main comment 1 of Reviewer #1.

We hope this clarifies the issue and demonstrates the reliability of our approach. Please let us know if you have any further questions or require additional details.

2) In Section 2.1, line 110, why did you choose to exclude column NO$_2$ has a climatology smaller than $1.43 \times 10^{15}$ molec/cm$^2$. How was this number determined?

The TROPOMI NO$_2$ retrievals were analyzed with a background noise cutoff, as suggested by previous studies (Qu et al., 2021; Tack et al., 2021), which is computed to be approximately $1.43 \times 10^{15}$ molecules/cm$^2$, based on a combination of both a constant term on the uncertainty as well as a term which scales with the total column loading impacting the uncertainty (in urban and industrial types of areas). The resulting solutions are inherently self-constrained within a specific range due to the physical constraints on the acceptable values of thermodynamic and chemical parameters used to drive the model. Additionally, the TROPOMI NO$_2$ retrievals were adjusted with a 40% increase to approximate findings from ground-based Max-DOAS studies suggesting that TROPOMI NO$_2$ values are underestimated. Conversely, a 40% decrease was applied to approximate studies indicating that co-emitted black carbon (BC), dust, and other absorbing aerosols cause TROPOMI NO$_2$ values to be overestimated. Furthermore, TROPOMI NO$_2$ retrievals were also evaluated with a lower error cutoff for background noise, set at ($1 \times 10^{15}$ molecules/cm$^2$). These constraints filter out TROPOMI NO$_2$ values that are not physically realistic given the in-situ environmental conditions. We err on the side of being conservative, and would rather only work with data that we consider sufficiently trustworthy. Consequently, the uncertainty in the computed emissions output is consistently observed to be narrower than the uncertainties in the TROPOMI NO$_2$ inputs.

3) In Section 3.1, Figure 5 plots the distribution of monthly $NO_x/NO_2$ over grids from different sources, how many facilities were counted for each emission source?

Thank you for your insightful comment regarding the number of facilities counted for each emission source in Section 3.1, Figure 5.

In our analysis, we utilized location data from pollutant sources provided by the Pollutant Discharge Permit Management Information Platform of the Ministry of Ecology and Environment to quantify the distribution of monthly $NO_x/NO_2$ over grids of five different sources. Specifically, the number of facilities counted for each emission source is as follows: 1) Power Plants: We included data from 163 power plants, which were identified as significant point sources of $NO_x$ emissions (Beirle et al., 2021). 2) Industrial Facilities: This category encompasses 109 steel and iron factories and 212 cement factories. 3) Residential and Commercial: This category includes emissions from 71 biomass burning facilities and 243 heat and production supply facilities, with data aggregated from urban and suburban areas (Jones et al., 2023).

Response Table 2. The number of facilities counted for each emission source in Section 3.1, Figure 5.

|  | Power Plants | Steel and Iron Factories | Cement Factories | Biomass Burning | Heat Production and supply |
|---|---|---|---|---|---|
| Number | 163 | 109 | 212 | 71 | 243 |

4) In Section 3.2, a key aspect to consider is the performance of the fitting process and the resulting errors in each parameter, as this is essential for assessing the reliability of the MCMFE framework. Given that the fitting incorporates all observations, does it remain unbiased across different months and grid cells?

We have responded the same comment made by Reviewer #1 (specific comment 4). To ensure clarity and thoroughness, we are providing the detailed response again:

In addition to the sensitivity analysis of the initial assumptions regarding $NO_x$ emissions, a robustness test of the mass-conserving flexible emissions inversion was also conducted. We have conducted a comprehensive sensitivity analysis to assess the robustness of our methods. This analysis is done in much greater depth and detail in a separate manuscript that which is currently under review in Remote Sensing of Environment (Lu et al., 2024). In this manuscript under review at the present time, we have assumed an additional case in which the TROPOMI $NO_2$ observed column values include uncertainties at the extreme upper and lower bounds of their ±40% uncertainty range respectively, and use these to re-compute the fitting factors

and subsequent emissions. When the TROPOMI $NO_2$ column values were adjusted, all related factors were simultaneously modified. This set of uncertainty simulations was uniformly applied as the $TO_{40\%}$ case, where the $NO_2$ columns were multiplied by random perturbations ranging from 0.6 to 1.4. The coefficients were refit using these new values from TROPOMI and the same values from both INTAC and meteorology over the entire domain included in this work. The results are provided in Response Figure 2 and Response Table 3.

The first point to note is that the refit values of the chemical and thermodynamic terms yield uncertainty ranges that are smaller than their respective perturbations, indicating that the fits are stable to the uncertainty perturbation. At the 40% uncertainty level, there is a reduction in short-term transport and an increase in long-term transport. This again is consistent with the uncertainty in the a priori emissions estimate. However, accounting for the net uncertainties (i.e., including the buffering effects of the chemical and thermodynamic terms with the more uncertain transport term) our overall findings demonstrate that the mass-conserving flexible emissions inversion method provides robust inversion results (as illustrated in Response Figure 3). This is especially so when compared to the traditional wind speed and concentration gradient method, as observed in the transport term being the least stable to the uncertainty perturbation.

[Figure]

Response Figure 2. The distributions of three key coefficients obtained from $TO_{0\%}$ and $TO_{40\%}$: a) $NO_x/NO_2$, b) Lifetime [hours], c) Transport [km]

Response Table 3. The 20th, 50th, 80th percentile ranges of three coefficients from TO0% and TO40%.

| NOx/NO2 | TO0% | TO40% |
|---|---|---|
| 20% | 2.76 | 2.63 |
| 50% | 6.30 | 6.08 |
| 80% | 14.6 | 13.4 |
| Lifetime (days) | TO0% | TO40% |
| 20% | 0.29 | 0.30 |
| 50% | 0.48 | 0.48 |
| 80% | 0.68 | 0.68 |
| Transport (km) | TO0% | TO40% |
| 20% | -323. | -403. |
| 50% | 18.1 | 24.0 |
| 80% | 351. | 416. |

[Figure]

Response Figure 3. The TROPOMI NO2 column data is perturbated by a random factor 40% herein called [TO40%] to

represent its range of uncertainty. The results are displayed for annual mean of emissions of a) $TO_{0\%}$, b) $TO_{40\%}$; and annual mean of error of c) $TO_{0\%}$, d) $TO_{40\%}$; and the annual mean of e) $(TO_{0\%}-TO_{40\%})/TO_{0\%}$

It was observed that 93% of the daily grid cells exhibited a ratio of $[(TO_{0\%}-TO_{40\%})/TO_{40\%}]$ within ±40%. The spatial and temporal 20th and 80th percentile ranges of $NO_x$ emissions for the $TO_{0\%}$ and $TO_{40\%}$ scenarios were 0.47 to 1.25 µg/m²/s and 0.45 to 1.25 µg/m²/s, respectively.

Overall, the day-by-day and grid-by-grid $NO_x$ emission ranges are quite similar in both cases, as illustrated in Response Figure 4. On a month-by-month basis, the deviations for nearly all months are within ±10% (February is excluded due to missing TROPOMI data in 2019 for region 2). January, October, and December exhibit the largest deviations, with $TO_{40\%}$ values consistently lower than $TO_{0\%}$ across all percentile ranges in January and October. Furthermore, the day-by-day spatial median values across different cases exhibit only slight variations. These findings indicate that changes in the driving factors ($\alpha_1$, $\alpha_2$ and $\alpha_3$) between the different $NO_2$ column loading scenarios are generally smooth and consistent, providing redundancy and being significantly influenced by the a priori emissions used in the fitting process. The constraints on the physically realistic values of $\alpha_1$ and $\alpha_2$, along with the constant use of INTAC, create a negative feedback loop affecting the relationship between $NO_2$ column changes and the final emissions products. This is consistent with the observed computed emissions and their differences.

[Figure]

**Response Figure 4**. a) The PDF of $NO_x$ emissions [μg/m²/s] over all individual days and grids of $TO_{0\%}$ and $TO_{40\%}$; b) The 20th, mean and 80th values across different months of $TO_{0\%}$ and $TO_{40\%}$; c) The time series of the spatial median values of $TO_{0\%}$ and $TO_{40\%}$ for whole year.

We have further identified Response Figure 3 and 4 as Figure S3 and Figure S4, with subsequent revisions in section 3.2. The specific changes in the manuscript can be found in our response to specific comment 4 of Reviewer #1.

5) In Section 3.4, Figure 8 plots have the unit of (Kton yr$^{-1}$ cell$^{-1}$), however, the size of a cell is not mentioned in the context.

Thank you for your observation regarding the units and cell sizes in Section 3.4, Figure 8. The units of (Kt yr$^{-1}$ cell$^{-1}$) represent the emissions per cell per year. We acknowledge that the size of each cell was not explicitly mentioned in the context. The actual grid is given to be 0.05°x0.05°, which means that the cells in our grid have varying areas due to differences in latitude and longitude, which affects the computation of total emissions of the whole year.

To address this, we have now included a detailed description of the grid cell sizes in the revised manuscript. "The grid cells are defined by a latitude-longitude grid with a resolution of 0.05°x0.05°, meaning that area of each cell varies with latitude. This variation is accounted for in emission calculations to ensure accurate representation of emissions per unit area."

6) Did the authors use MEIC emissions data or assume monthly invariant emissions?

We have reorganized the paragraph of section 2.3 in the following way to be clearer. The fact is that the MEIC emissions data themselves assume monthly invariant emissions, so we decided to not impose additional variation. Please see the details below:

"The assumptions regarding NO$_x$ emission datasets in the initial step applied are harmonized using multi-source heterogenous data, developed by the MEIC (Multi-resolution Emission Inventory for China) team (Huang et al., 2012, 2021; Kang et al., 2016; Liu et al., 2016; Zheng et al., 2021; Zhou et al., 2017, 2021), in collaboration with various scientific research institutions. This dataset is referred to as the high-resolution INTegrated emission inventory of Air pollutants for China (INTAC), which is highlighted in purple in Figure 3. The original INTAC emissions are quantified in units of Mg/grid/month, with a temporal resolution of one month and a spatial resolution of 0.1° x 0.1°, for the year 2017. It is important to note that higher resolution inventories, such as the 1-km resolution inventory developed by the MEIC team at Tsinghua University, is also available (Zheng et al., 2021). However, the 1-km inventory is for 2013, and the INTAC inventory we utilized offers the highest resolution available which is the closest temporal match to 2019 TROPOMI data. This dataset covers mainland China and includes emissions from eight sectors: power, industry, residential, transportation, agriculture, solvent use, shipping, and open biomass burning(Wu et al., 2024). To align the resolution of the original INTAC Inventory with that of TROPOMI grids, we undertake several processing steps: 1) The units are converted from Mg/grid/month to $\mu g/m^2/s$ as the first step, due to the varying areas of each longitude-latitude grid. 2) Next, the INTAC inventory is adjusted to a 0.05°x0.05° grid using the nearest neighbor method. 3) Finally, we assume that the monthly emissions remain constant on a day-to-day basis. To ensure that the values used do not fall within the error range of the TROPOMI sensor (i.e., noise), values below 0.2 $\mu g/m^2/s$ are designated as NaN and are not considered further in this study."

Minor comments:

1) Line 215 and line 338, "densitiy" should be "density".

Thank you, we have made this modification.

2) Line 127, "boarder" should be "border".

Thank you, it has been modified.

3) Line 232, typo error "heat productin and supportion". Line 234, "form" should be "from".

Thank you for helping us improve this!

4) Line 262, typo error "reated".

Thank you, it has been modified.

5) Line 283, typo error "Qingdai".

Thank you for catching this. We have fixed it.

6) Line 363, typo error "differnces".

Thank you again.

7) Line 289 and 424: However -> Moreover

Thank you, it has been modified.

8) Line 360, 380 and 388: differences -> the differences

This has been updated.

9) Line 18: Kton/year -> Kt/year. Please ensure that units are consistently presented and aligns with standard publication practices.

All units are not consistent. Thank you for your help with our communication.

10) Please use hyphen (-) in the words separated by two lines or change the format.

Thank you, it has been modified.